# Physics-aware differentiable design of magnetically actuated kirigami for shape morphing

Liwei Wang [1], Yilong Chang[2], Shuai Wu [2], Ruike Renee Zhao [2] & Wei Chen [1] ✉

Shape morphing that transforms morphologies in response to stimuli is crucial for future multifunctional systems. While kirigami holds great promise in enhancing shape-morphing, existing designs primarily focus on kinematics and overlook the underlying physics. This study introduces a differentiable inverse design framework that considers the physical interplay between geometry, materials, and stimuli of active kirigami, made by soft material embedded with magnetic particles, to realize target shape-morphing upon magnetic excitation. We achieve this by combining differentiable kinematics and energy models into a constrained optimization, simultaneously designing the cuts and magnetization orientations to ensure kinematic and physical feasibility. Complex kirigami designs are obtained automatically with unparalleled efficiency, which can be remotely controlled to morph into intricate target shapes and even multiple states. The proposed framework can be extended to accommodate various active systems, bridging geometry and physics to push the frontiers in shape-morphing applications, like flexible electronics and minimally invasive surgery.

Shape-morphing systems that can undergo morphological changes in response to external stimuli have great potential for a wide range of applications, such as soft robotics[1–7], minimally invasive surgery[8–10], and flexible electronics[11,12]. These shape morphing applications typically require non-uniform and large deformation in the structures to realize intricate functional shapes, which is challenging to realize with ordinary materials. In contrast, inspired by the ancient art of paper cutting, kirigami introduces cuts to relax the continuum constraints in materials, allowing for significant spatial variation of the deformation within. The resulting versatile deployment processes make kirigami a promising choice in shape morphing[13–15]. Moreover, the integration of stimuli-responsive materials with the inherent kinematic flexibility offered by kirigami has led to the emergence of active kirigami systems. They enable the realization of intricate but controllable dynamic shape changes of kirigami that were previously unattainable[14,16–23].

Despite its great potential, kirigami has been largely focusing on regular periodic or heuristic cuttings, where cutting patterns are typically predetermined with hand-picked parameters rather than pursuing on-demand inverse design approaches[13]. Several studies have explored inverse design methods, mainly focusing on parameter optimization, such as hinge widths and panel aspect ratios[24,25]. Moreover, these studies have primarily been conducted for the unit cell of periodic kirigami configurations, which will restrict the flexibility in realizing complex shapes. To further harness kirigami's shape-morphing capabilities for intricate functionalities, a recent inverse design framework has successfully relaxed the periodicity requirement in kirigami, creating optimized aperiodic cutting patterns to realize versatile deployed shapes[26]. This approach has been further extended to encompass various types of kirigami, addressing complex kinematic requirements such as compact reconfigurable designs[27], and accommodating different topologies[28]. However, to the best of the authors'

[1]Department of Mechanical Engineering, Northwestern University, Evanston, IL 60208, USA. [2]Department of Mechanical Engineering, Stanford University, Stanford, CA 94305, USA. ✉e-mail: weichen@northwestern.edu

knowledge, existing inverse design processes for shape-morphing kirigami, including the aforementioned recent works, primarily focus on the design of geometries or kinematics but do not explicitly consider the physics[13,29–31]. By physics, we mean the fundamental laws and principles underlying the forces, energy, and other physical interactions governing the deployment or actuation process. Consequently, most existing kirigami designs fail to address the essential physical feasibility and rely on simple mechanical manipulation (mostly by hand) to drive the shape-morphing process, which is impractical in real applications. There are a few physics-aware exceptions, most of which only consider physics in post-analysis after completing the design process[26,27,32] or assembling precomputed unit cells with weak interactions[33–37]. In instances where physical simulations are directly integrated into the inverse design loop, they often incorporate full finite element analysis (FEA)[16,19,20,22,23,38,39] and primarily account for mechanical loading only. Moreover, these design processes frequently resort to brute-force parameter sweeping or gradient-free heuristic optimizers. The applicability of these optimizers, combined with the resource-intensive nested FEA, confines the scope of these designs to low-dimensional problems with simple constraints, such as periodic or unit-cell designs. Overall, there is a lack of an efficient and flexible physics-aware inverse design framework, which hampers the integration of kirigami with stimuli-responsive materials to realize more complex and practical applications.

The key barrier in designing physics-aware active kirigami is to incorporate the complex interplay between geometry, materials, and external stimuli in an iterative, automated design process. Furthermore, simulating the deployment process is often time-consuming and non-differentiable, without the analytical gradient of the design objective required to effectively navigate in a high-dimensional design space. As a result, the consideration of multiple physics often appears incompatible with high design flexibility and efficiency. This study addresses these issues and demonstrates how to explicitly and yet efficiently incorporate both geometry and physics into kirigami design for active shape morphing via differentiable modeling and gradient-based optimization. It aims to fill the existing knowledge gap in physics-aware kirigami design, enabling a better understanding of the design principles and efficient optimization processes. We focus on magnetically actuated kirigami made of hard-magnetic soft material as the illustrative case. Magnetic actuation offers the advantage of performing tasks remotely in confined and enclosed spaces, making it particularly valuable for many shape-morphing applications mentioned earlier. It has demonstrated promise in fields such as surgical robotics and flexible electronics[40], where precise control, intricate behaviors, and rapid customization are required. Given the practical significance of these applications, the need for an improved design approach that considers physics while maintaining high flexibility and efficiency becomes even more critical. Meanwhile, magnetic actuation presents unique challenges due to its position- and deformation-dependent magnetic potential[40–42]. It involves complex interactions between geometries, materials, and external stimuli, which are also encountered in other types of actuation such as thermal load[23], humidity[43], and pH[8]. Therefore, the design principles and insights obtained in this study have broad applicability across various designs actuated by different physics. Specifically, we propose an energy-based differentiable inverse design framework for magnetically actuated kirigami, explicitly incorporating physics into the design process. This approach allows simultaneous optimization of geometry and active materials to achieve complex shape-morphing behaviors, including multi-state designs that can freely transform into different stable configurations under different magnetic stimuli. It demonstrates superior efficiency, effectiveness and flexibility in quickly responding to new design scenarios for solutions with both kinematic and physical feasibility, unlocking design possibilities that were previously unachievable.

## Results

### Physics-aware differentiable design

While our method can be applied to various kirigami patterns, we have chosen to focus on the quadrilateral kirigami pattern for ease of illustration. As illustrated in Fig. 1a, our design process begins with a compact quadrilateral kirigami consisting of a repeating unit cell of four square panels. The panels are connected by hinges at the nodes to enforce mutual kinematic constraints, only allowing each pair of connected panels to counter-rotate and uniformly morph the overall configuration into a squared deployed shape. To achieve a kinematically admissible path between the compact and deployed states, it is important to ensure geometrical compatibility between the panels, as depicted in Fig. 1b. For instance, edges overlapped in the compact state must have equal lengths, and the panel angles around the center node of a compacted unit cell must add up to $2\pi$. Beyond ensuring geometrical compatibility, how to actuate the kirigami into its deployed state is also a crucial aspect to consider in real-world applications, which is often overlooked in existing designs. In our design, we utilize magnetic torque as the actuation to enable remote and active control of the kirigami. This is achieved by uniformly dispersing hard-magnetic particles with programmed magnetization within the polymer matrix of each panel through a direct ink writing (DIW) printing method[44,45] (See Methods for material and printing details), as illustrated in Fig. 1a. A uniform magnetic field $\mathbf{B} = B\mathbf{e}_B$ of magnitude $B$ and direction $\mathbf{e}_B$ will then impart distributed magnetic torques in kirigami to achieve shape morphing (Fig. 1a). For the $i$th panel with magnetization vector $\mathbf{M}_i = M\mathbf{e}_i$ of magnitude $M$ and direction $\mathbf{e}_i$, the induced magnetic torque can be computed as

$$\boldsymbol{\tau}_m = bMA_i\mathbf{e}_i \times \mathbf{B}, \qquad (1)$$

where $b$ and $A_i$ are the thickness and area of the panel, respectively. The positive direction of x- and y-components for both $\mathbf{e}_B$ and $\mathbf{e}_i$ is defined to be aligned with the axes shown in Fig.1a. The magnetization direction $\mathbf{e}_i$ of each panel should be carefully designed so that the induced magnetic torque can rotate the panel into the desired orientation upon the applied magnetic field. Specifically, the deployed configuration of the kirigami should satisfy the physical equilibrium between magnetic torques and mechanical forces, as shown in Fig. 1c. In a conventional kirigami design without considering the physics, this equilibrium condition is usually not satisfied, and thus the design is often physically infeasible. Our goal is to develop a fully automated inverse design approach so that for various given target deployed shapes, the kirigami cutting and magnetization of each panel can be rapidly obtained to achieve the desired shapes after actuation while ensuring both geometrical and physical feasibility.

Since periodic cuttings impose significant restrictions on the achievable deployed shapes, we turn to general kirigami designs with aperiodic cuttings. We begin by conformally mapping the deployed configuration of a regular kirigami (Fig. 1a) into the desired target shapes (Fig. 1d) using Schwarz-Christoffel mapping[46]. The magnetization orientation of each panel is adjusted accordingly by the average angle change of vectors connecting the center and the four nodes. The resulting kirigami design may no longer meet the geometrical compatibility requirements, and the deployed state is usually not in magneto-elastic equilibrium. As a result, although the deployed shape closely approximates the target, it is impossible to retrieve a compact kirigami design that can morph into the deployed shape in a geometrically and physically feasible manner (Fig. 1d). To address these issues, we conduct a constrained optimization on the mapped deployed state (Fig. 1f) to optimize both the cutting and magnetization orientation for a geometrically and physically feasible deployed state (Fig. 1e), from which the compact design can be easily identified by direct contraction.

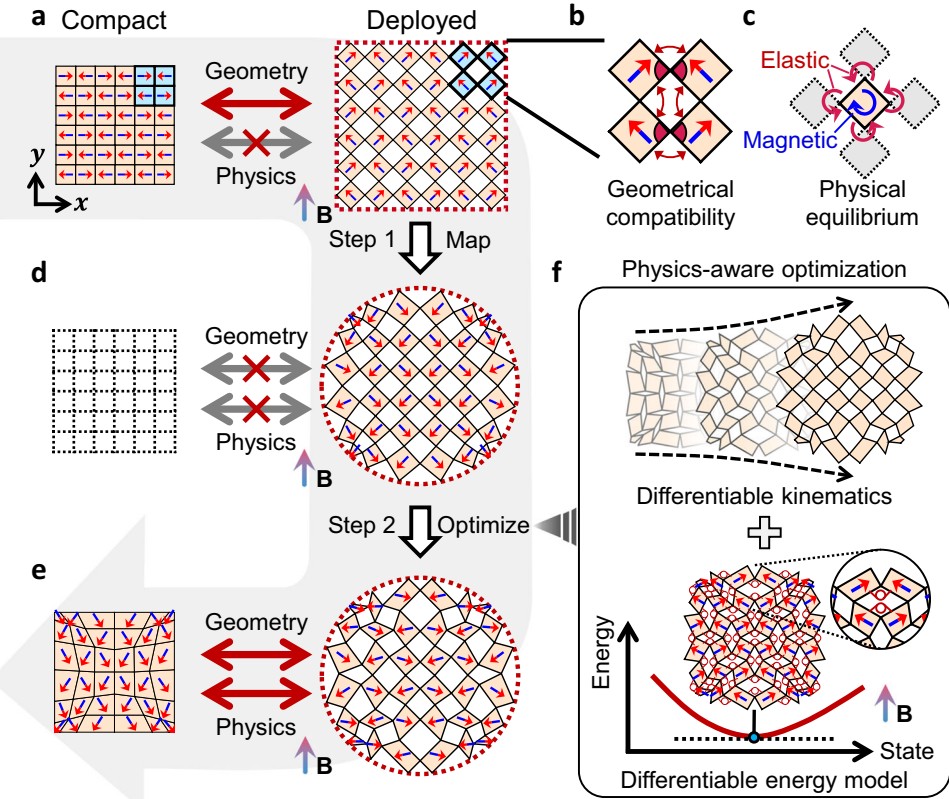

**Fig. 1 | Schematic diagram of the physics-aware differentiable design of kirigami. a** Compact state (left) of a regular quadrilateral kirigami and its deployed state (right) transformed from the regular compact kirigami following a geometrically feasible path. The repeated four-panel unit cell is shaded in blue, where the arrow shows the magnetization orientation of each panel. **b** Geometrical compatibility requirements for a four-panel cell in edges (equal-length pairs connected by red curved arrows) and angles around the center node (red sectors). **c** Torque induced by the applied magnetic field (blue curved arrow) and the mechanical forces/torques induced by deformation (red curved arrows) are required to be in equilibrium for a physically feasible deployed state. **d** The target deployed state (right) is conformally mapped from the deployed kirigami in Fig. 1a to achieve

predetermined shapes (red dashed circular contour as an example) under the magnetic field **B**. However, there is no compact state (left) that can follow a geometrically and physically feasible path to this mapped deployed state. **e** Compact state (left) and deployed state (right) of the kirigami optimized from the conformally mapped design in Fig.1d, satisfying both physical equilibrium and geometrical compatibility. **f** This physics-aware optimization integrates two models, the differentiable kinematic model (top) and the differentiable energy model (bottom) under a fixed external magnetic field. The enlarged inset shows the equivalent nonlinear springs for kirigami hinges. The optimization process achieves physical equilibrium by ensuring minimal energy of the deployed kirigami indicated by zero gradients.

It is important to acknowledge that although optimization-based inverse design methods have been proposed in the literature to achieve geometrical compatibility[26–28], on which our method is built, they are unable to incorporate physical equilibrium requirements into the design process. This is due to the complex interactions among the panels and the strong elastic-magnetic coupling, as demonstrated in Fig. 1c. The resulting lack of gradient information of the design objective precludes the use of efficient and effective gradient-based solvers, and the computational cost for multi-physics simulation is prohibitive for iterative design (e.g., a typical genetic-algorithm-based search usually requires thousands of evaluations[38,47] and may take days or even weeks in our case). Furthermore, the compact and deployed states are intertwined to determine the physical equilibrium. The equilibrium is thus design-dependent and changes iteratively throughout the design process. This results in a dynamic optimization problem that is notoriously difficult to solve. To overcome these challenges, we first develop differentiable kinematic and energy models of kirigami (Fig. 1f). Among all kinematically admissible configurations, only the one corresponding to the total energy minimum can achieve physical equilibrium and remain stable. Ideally, we want this minimal-energy configuration to be the designed deployed state so that the kirigami can transform into and retain the target shape under a given stimulus. With the differentiable models, we can easily integrate this minimal-energy requirement into a constrained

optimization framework with an analytical gradient to enable automatic and efficient solutions (Supplementary Note S6).

Specifically, assuming an external magnetic field aligns along the vertical direction, to achieve compatible counter-rotation in a four-panel basic cell (Fig. 2a, b), the horizontal components (x-axis) of magnetization in each panel (Fig. 2b) should have the sign indicated in Fig. 2a. Hence, we only need to use the vertical component (y-component) $d_i$ of a unit magnetization vector to determine the orientation of the magnetization panel, which is combined with the coordinates $(x_i, y_i)$ of the nodes as the design variables (Fig. 2c).

To ensure the geometrical feasibility of the deployed design, we apply constraints to the edges and angles, as described in Supplementary Note S1 and illustrated in Supplementary Fig. S1a–c. Additionally, we incorporate constraints on the deployed contour to keep it aligned with the target shape (Supplementary Note S1.4 and Supplementary Fig. S1d). An optional constraint is introduced to regularize the shape and aspect ratio of the compact state (Supplementary Note S1.5 and Supplementary Fig. S1e). To prevent overall rigid-body rotation when subjected to external excitation, the kirigami design is constrained to be symmetric along the vertical axis. This symmetry requirement ensures that the kirigami maintains net magnetization to be zero or along the external magnetic field throughout the reconfiguration process. While these geometrical constraints ensure a geometrically feasible compact kirigami (Fig. 2d) can always be retrieved

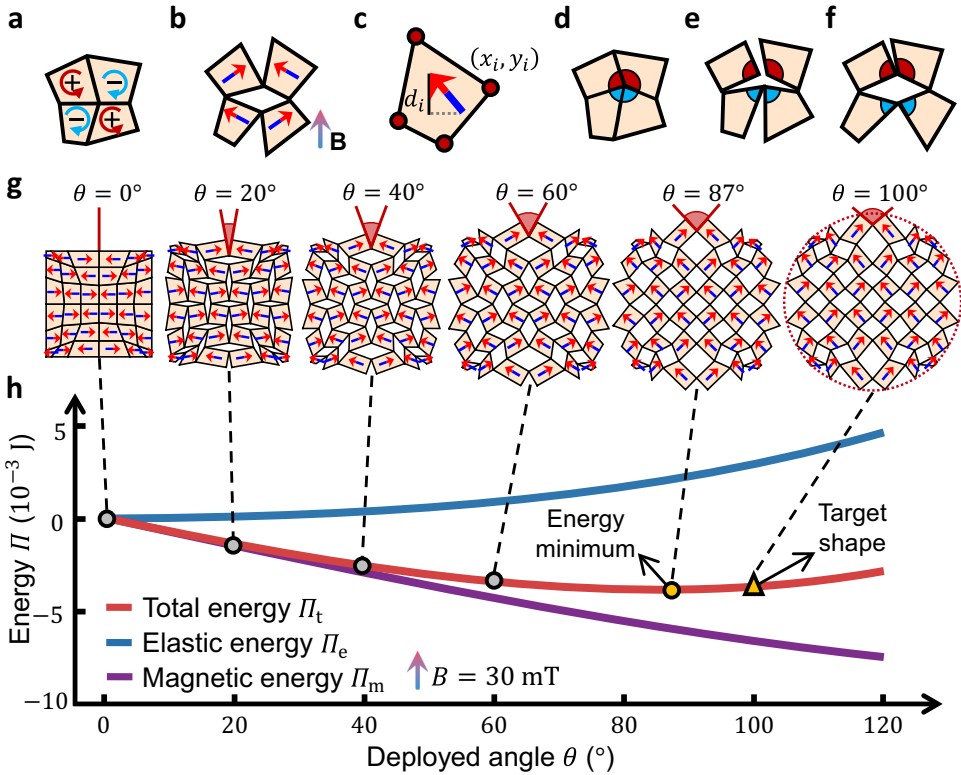

**Fig. 2 | Optimization problem setting and energy model. a** Kinematically compatible rotation direction of each panel in a basic cell, with positive and negative rotation marked by red and blue arrows, respectively. The sign of the horizontal component of magnetization is marked in each panel to achieve compatible counter-rotation. **b** Compatible counter-rotation of panels can be achieved under magnetic excitation by dispersing magnetic particles with horizontal magnetization components corresponding to signs marked in Fig. 2a. **c** Nodal coordinates and vertical components of unit magnetization vectors are used as design variables. **d** Panels with angle pairs (marked in the same colors) violating rigid-deployable constraints will encounter **e** geometrical incompatibility in the deployment process, leading to (**f**) geometrical frustration in the deployed state. **g** Configurations of a rigid-deployable kirigami corresponding to different deployed angles marked in red. **h** Elastic, magnetic, and total energies for configurations of different deployed angles, under the same constant external magnetic field $B = 30$ mT. Markers represent kinematically admissible configurations corresponding to those in Fig. 2g. Without designing the panel magnetization orientation (aligning horizontally in the compact state), the kirigami can only transform into the minimal-energy configuration with $\theta = 87°$ (marked by the yellow dot), failing to achieve the target deployed shape with $\theta = 100°$ (marked by the yellow triangle). Source data are provided as a Source Data file.

for the design deployed kirigami (Fig. 2f), they do not guarantee a kinematically admissible morphing path between the two states (Fig. 2e). Geometrical frustration can still occur in the kirigami, making it difficult to obtain an analytical energy model. To address this, we add an extra constraint on the angles around each of the rotating hinges (marked in the same color in Fig.2d-f). Each pair of these angles (red or blue color) should sum up to $\pi$ and achieve a straight cutting[27]. A kirigami satisfying this constraint is called rigid-deployable. It can be considered as a mechanism with a single degree of freedom (DOF), whose configuration can be fully determined by the deployed angle $\theta$ as shown in Fig. 2g.

To incorporate physics into the optimization process, we have developed an analytical model to describe the total energy $\Pi_t$ of a given rigid-deployable kirigami, which consists of elastic energy $\Pi_e$ and magnetic energy (potential) $\Pi_m$, as shown in Fig. 2h. Observing that the panel only has negligible deformation, we assume the panel to be rigid and utilize a modified hyper-elastic beam model to describe the elastic energy in a bending hinge induced by the counter-rotation between a pair of panels, expressed as an analytical function of the rotation angle (see Supplementary Note S4 and Supplementary Fig. S3)[48]. It can be considered an equivalent nonlinear spring shown in Fig. 1f. Then, given a kirigami configuration, we can obtain the total elastic energy $\Pi_t$ from the rotated angles of all the hinges. Meanwhile, the magnetic energy potential $\Pi_m$ is calculated by summing up the

potential of all $n_p$ panel as

$$\Pi_m = \sum_{i=1}^{n_p} -bMA_i\mathbf{e}_i \cdot \mathbf{B}. \tag{2}$$

Note that, the magnetic interaction between panels is very small and it has a negligible effect on the magnetic actuation of the printed kirigami, which is thus ignored here. Consequently, the total energy can be obtained via

$$\Pi_t = \Pi_e + \Pi_m, \tag{3}$$

which is a function of the deployed angle (Fig. 2h) for a fixed external magnetic field once the kirigami design is given. The stable deployed state thus corresponds to the deployed angle with the lowest total energy (yellow dot in Fig. 2h). It is noted in Fig. 2h that, there is a complicated relation between geometry (deployed angle) and the energies. As a result, when the physics or panel magnetization orientation is not taken into account in the design, the kirigami tends to deviate from physical equilibrium in the intended deployed shape (yellow triangle in Fig. 2h), leading to a deployed shape that differs from the target.

To realize the co-design of geometry and magnetization orientation, we further develop a differentiable kinematic analysis, from

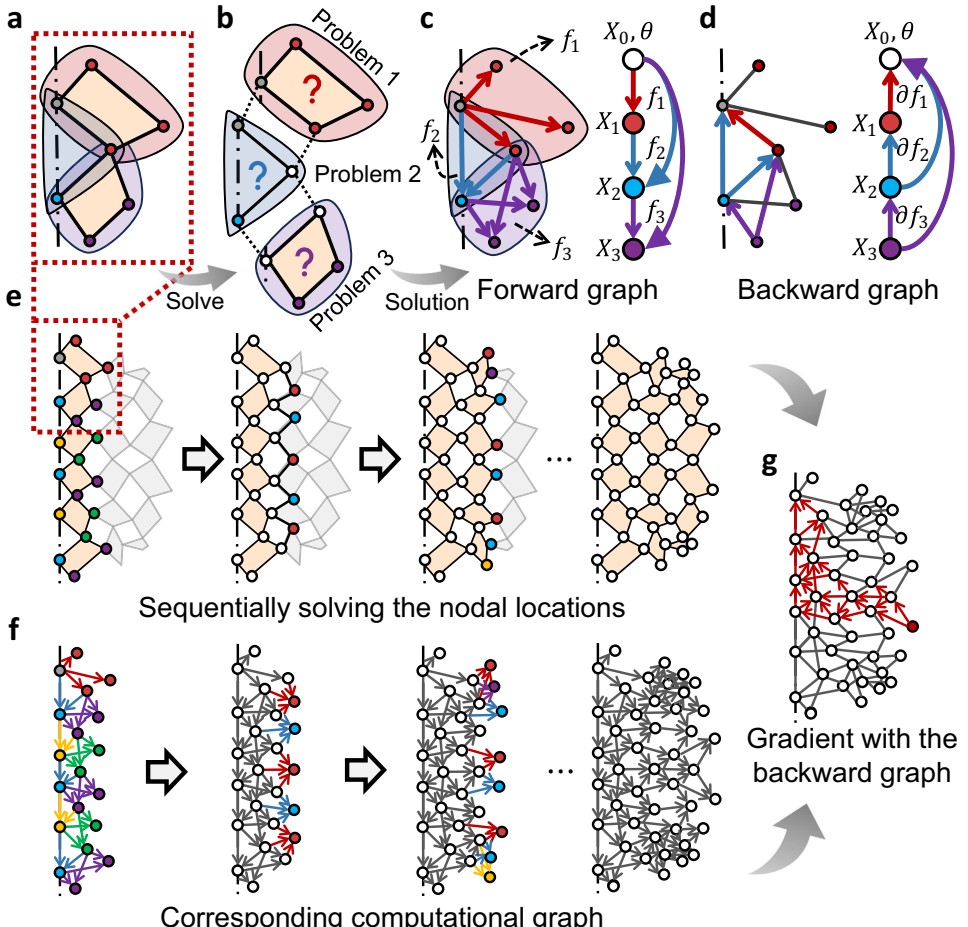

**Fig. 3 | Differentiable kinematic model. a** Two exemplified panels in kinematic analysis corresponding to the red dashed box in the subfigure below. The gray dot is fixed and the locations of the remaining colored nodes are solved, which can be divided into three basic geometry problems indicated by shades in different colors. **b** Details of three basic geometry problems to be solved sequentially. In each problem, nodes in colors can be analytically solved based on the fixed nodes (gray) and already-solved nodes (white). The solution of each problem can be considered as a transformation from known nodes to nodes to be solved. **c** Extracted forward computational graph (left) from Fig. 3b by composing the transformation in sequence. Different types of transformations are marked by arrows in different colors, pointing from preceding nodes to the updated nodes. Shaded regions represent geometry problems, from which the analytical expression of the transformations $f_i$ are obtained. A simplified graph is shown on the right. $X_0$ and $\theta$ are the original node locations and deployed angle, respectively. $X_i$ is the set of updated nodes obtained from each transformation, corresponding to nodes of the same color in the original graph. **d** Backward computational graph to calculate the gradient of locations for the node of interest by composing the gradient of individual transformation $\partial f_i$ via chain rules. **e** Sequential solving process to obtain all nodal locations. Only the right half is shown here as the left half is obtained by mirror symmetry on the center line. **f** Forward computational graph corresponding to sequential steps in Fig. 3e. The analytical transformations in each step are marked by arrows in different colors. Nodes that have been updated in previous steps are colored in white. **g** Backward computational graph (marked in red) to calculate the gradient of a given node.

which an analytical expression for the total energy, i.e., $\Pi_t(\theta)$, can be obtained to facilitate later simulation and design optimization. Given any change in the deployed angle, we solve for the corresponding kirigami configuration (kinematic analysis) sequentially as shown in Fig. 3. Specifically, calculating nodal locations of kirigami panels (Fig. 3a) involves a series of simple geometrical problems (marked by shaded regions in Fig. 3b), in which we solve for the unknown nodes (colored nodes) based on their constraints with the nodes obtained in preceding problems. Solutions for nodal locations in these basic problems can be analytically derived, which can be viewed as a series of analytical transformations $f_i$ (marked by colored arrows in Fig. 3c) from the preceding nodes to unsolved nodes ($X_0$ to $X_3$ and so on). By composing these transformations in sequence, they form a directed computational graph for forward analysis (Fig. 3c). Using chain rules, we can obtain the gradient values of node locations of any nodes by tracing the computational graph backward and multiplying the gradient of basic transformation $\partial f_i$ in order (Fig. 3d). We apply this kinematic analysis to the whole kirigami, iterating nodes in every

column of panels or voids (Fig. 3e) and sequentially composing the transformations together to get the analytical expression for the kirigami configuration. This analytical expression can be represented by a directed computational graph in Fig. 3f. Note that, due to the inherent regularity of cutting topology, i.e., the interconnections between panels, the solutions in each column only involve a limited set of transformation types (indicated by arrows in different colors in Fig. 3f). Using the chain rule again, we can readily calculate the analytical gradient of any nodal locations with respect to the deployed angle $\theta$ and initial node locations $X_0$ (Fig. 3g). By integrating these analytical kinematics into the previous energy model, we can derive analytical expressions for the total energy $\Pi_t(\theta)$ and its gradient. A more comprehensive description of the kinematics and energy models is included in Supplementary Note S3 and S5, respectively.

As stated earlier, a physical equilibrium state should correspond to a minimum of the total energy, which implies a zero energy gradient with respect to the deployed angle. This observation has two important uses in kirigami design. Firstly, it allows for a simple gradient-based

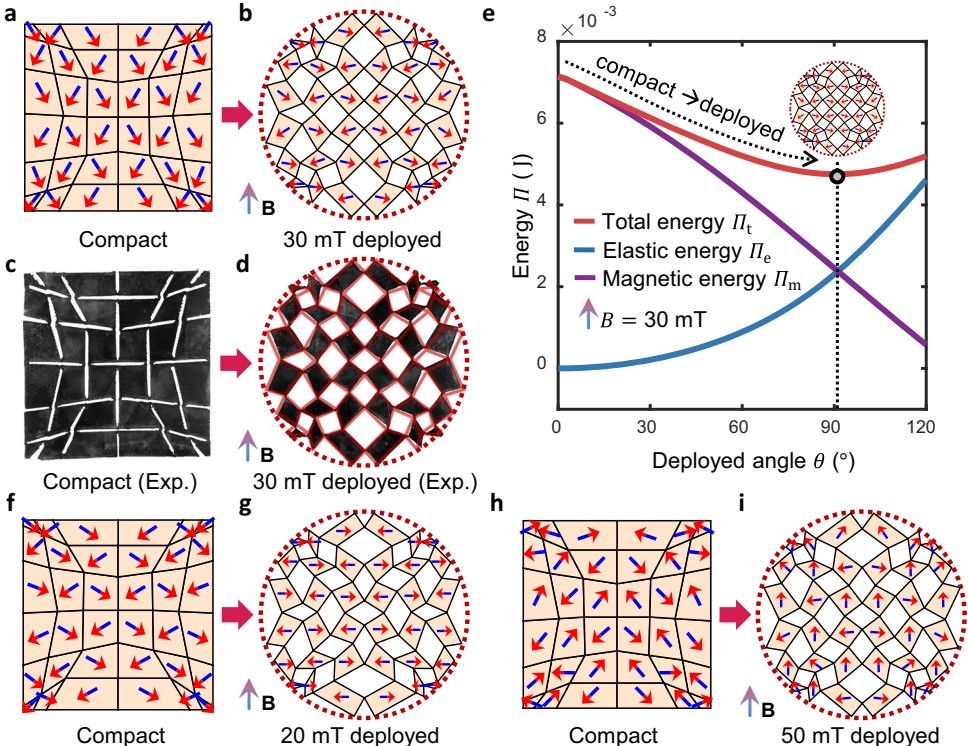

**Fig. 4 | Optimized design results to achieve a circular deployed shape. a, b** Show simulated compact and deployed states of optimized kirigami designed for $B = 30$mT, respectively. All deployed shapes in this figure are rescaled to a similar size as their compact shapes for a better layout. Unscaled results are shown in Supplementary Movie S1-2. **c, d** are experimental (Exp.) results corresponding to those in Fig. 4a, b, respectively. The transparent red lines mark the configuration of the simulated design in Fig. 4b. **e** Energy analysis of the designed kirigami in Fig. 4a, under the constant field $B = 30$mT. The deployed state shown in Fig. 4b corresponds to the lowest energy state marked by the vertical dashed line. Upon the given constant excitation, the compact state will have a higher total energy and will transform to the energy minimum (designed deployed state), as marked by the dashed arrow. Source data are provided as a Source Data file. **f, g** Show simulated compact and deployed states of optimized kirigami designed for $B = 20$mT, respectively. **h** and **i** show simulated compact and deployed states of optimized kirigami designed for $B = 50$mT, respectively.

search to find the stable deployed state with the lowest energy, which is proved to be more efficient than finite element methods (within seconds compared to ~10 min) with approximately the same accuracy (see Supplementary Note S3, S5.4, and Supplementary Fig. S4–6). We will utilize this energy-based simulation to validate designs in this study. Secondly, the logarithm of the squared energy gradient,

$$\min_{x_i, y_i, d_i} \mathcal{F} = \ln\left[\left(\frac{\partial \Pi_t}{\partial \theta}\right)^2 + \delta\right], \qquad (4)$$

is used as the objective function in the design to ensure physical equilibrium, where $\delta = 2 \times 10^{-16}$ is a small constant to avoid singularity issues. This objective solely depends on the deployed state and therefore avoids the original dynamic optimization problem caused by the coupling with the compact state. Since both the objective and constraints are differentiable, we can efficiently find the optimal design using sequential quadratic programming (SQP), a widely adopted gradient-based approach for solving nonlinear, constrained optimization problems[49]. It should be noted that, since both types of energies are linearly related to the thickness of the kirigami, the thickness value will not influence the design outcomes or the physical equilibrium of the system. Therefore, during the design process, we assume the thickness to be large enough in real applications to keep the kirigami always in-plane. Supplementary Note S6 provides further insights and details regarding the optimization process.

### Shape morphing designs to achieve circular deployed shapes
To showcase our approach, we begin by designing a square compact kirigami that transforms into a circular shape when exposed to an external magnetic field, with a positive vertical direction (aligning along the y-axis) and magnitude $B = 30$mT. The kirigami is assumed to have a thickness of 0.8mm, composed of silicone-based resin with 5 μm neodymium-iron-boron (NdFeB) particles embedded, with a shear modulus of 300kPa, a Poisson's ratio of 0.495, and magnetic moment density of the composite $M = 70$kA/m. For ease of illustration and consideration of manufacturability, we choose to stack $3 \times 3$ four-panel basic cells in the kirigami, connected by cuboid hinges of size 1.2mm × 1.6mm × 0.8mm. We also include the constraint on the compacted state to ensure a square shape. The proposed differentiable inverse design method allows for efficient optimization of the kirigami, taking only a few minutes as opposed to days or even weeks required when using a heuristic optimizer integrated with non-differentiable simulation. The design results are shown in Fig. 4a-b and Supplementary Movie S1.

Compared with the regular grid-like cutting in periodic designs (Fig. 1a), the optimized cutting is non-uniform, with panels on the outer boundary significantly distorted. Panels that tend to move outwards in the deployment are elongated, e.g., segments centered at the four boundaries, while the panels at the corners are greatly compressed to achieve the circular deployed shape. This is a typical way for kirigami to utilize the discontinuity introduced by the cutting to redistribute the deformation for better shape matching[13]. Despite the highly non-uniform shapes, panels are still compatible and rigid-deployable, due to the imposed geometrical constraints. The optimized magnetization orientation is also distinct from that in the periodic (Fig. 1a) or initial mapped designs (Fig. 1d). In the compact state, the magnetization orientation is more aligned in the opposite direction of the external magnetic field, while in the deployed state, the

orientation is closer to the perpendicular direction. This leads to a significant decrease in magnetic potential energy that compensates for the increase in elastic energy during the deployment process, as demonstrated by the energy analysis in Fig. 4e. It means that a larger magnetic torque is induced in the deployed state to counteract the elastic forces and maintain the deployed shape. As a result, the deployed state corresponds to the state of minimal energy in physical equilibrium (Fig. 4e), in contrast to the unstable target deployed shape in Fig. 2g, h without the magnetization orientation design. To further validate our findings, we conducted experiments on 3D-printed kirigami, as depicted in Fig. 4c, d and Supplementary Movie S2. The details for the manufacturing and experiment can be found in the Methods section. From Fig. 4d, it can be observed that the deployed angle is almost the same for simulated and experimental designs. Despite a small discrepancy due to manufacturing errors and frictions, the experimental results overall demonstrate a close agreement with the simulated designs in achieving the desired circular deployed shape.

### Shape morphing designs under different magnetic field magnitudes

To investigate the effect of the external magnetic field, we designed another two kirigami structures to achieve the same circular deployed shapes but under different magnitudes of $B$. When subjected to a weaker magnetic field of $B = 20$mT, as shown in Fig. 4f, g, the optimized cutting is modified in such a way that the deployed angle is smaller than that in the design for $B = 30$mT (Fig. 4b, Supplementary Movie S1), resulting in lower elastic energy and thus smaller elastic forces in hinges. Meanwhile, all panels have their magnetization orientation aligned perfectly perpendicular to the external field in the deployed state, maximizing the induced magnetic torque. The reduction in elastic forces resulting from the altered cutting and the increase in magnetic excitation induced by the change in magnetization orientations combine to maintain equilibrium in the same deployed shape, even in a much weaker field. On the other hand, when exposed to a much stronger field with $B = 50$mT, the magnetic potential is more than enough to induce torques compensating for the elastic forces. As a result, the optimized design, as shown in Fig. 4h, i and Supplementary Movie S1, has more panels with their magnetization orientation aligned with the external magnetic field in the deployed state, leading to a decrease in magnetic potential and torques, and thus maintaining a physically stable deployed state. These findings suggest that the interplay among geometrical cutting, magnetization orientations in the panels, and the magnetic fields plays a crucial role in achieving target deployed shapes in physical equilibrium. Our approach can take into account this interplay and thus enable the co-design of different entities for optimal performance.

### Shape morphing designs to achieve various deployed shapes

While many existing designs rely on trial-and-error or heuristic designs, the proposed method offers the flexibility to accommodate direct inverse design for various complex deployed shapes, as shown in Fig. 5 and Supplementary Movie S3. In these cases, we use the same upward magnetic field with a magnitude of $B = 35$mT. The compact shapes are still constrained to be rectangular, but their aspect ratios are relaxed to be freely changed by the optimization to further increase flexibility. The results demonstrate the success of the proposed method in achieving deployed shapes that precisely match the given targets, even when the target shapes exhibit significant changes in aspect ratios between states (Fig. 5a) or drastic variations in curvatures (e.g., Fig. 5b, c, e, f). In particular, we perform experiments on the 3D-printed goblet-like design (Fig. 5g and Supplementary Movie S4), whose deployed angle and overall deployed state (Fig. 5g) match well with the design target and simulation (Fig. 5f). Similar to our previous designs for a circular deployed shape, the optimized cuttings in all

these cases here lead to non-uniform panels. Panels that expand outward in the deployed state are elongated or enlarged, while panels that contract inward are compressed. This non-uniformity becomes more pronounced when there is a substantial change in aspect ratio (Fig. 5a) or boundary curvatures (e.g., Fig. 5b, c, e, f) during deployment, enabling spatially varying deformation for target shape morphing.

It is interesting to note that a larger change in shapes/deployed angles between the two states results in panels having magnetization in deployed states more perpendicular to the external fields (as seen in Fig. 5c, 5f). In the same deployed kirigami, panels with smaller sizes or larger rotation angles have magnetization orientations more perpendicular to the external field, while panels with larger sizes or smaller rotation angles have magnetization orientations more aligned with the external field. For instance, in Fig. 5f, the lower half panels have smaller sizes compared to panels in the upper half, which have magnetization more perpendicular to the external field. Consequently, both parts contribute similar magnitudes of magnetic torques to balance the competing magnetic and elastic forces. These observations underscore the intimate connection between designs on magnetization and geometry in achieving physically stable deployed states, further validating the effectiveness of our proposed method.

### Two-way contractible designs with target deployed shapes

All the results discussed thus far have focused on designs that transform only between a single compact state and a deployed state. However, our proposed method can also design kirigami that morphs into multiple equilibrium states when subjected to different magnetic fields. We illustrate this capability in Fig. 6a, where the kirigami undergoes three distinct equilibrium states: zero, positive, and negative states. In the zero state, no magnetic field is applied, and the deployed kirigami is considered as the rest configuration (un-actuated and undeformed), which is different from previous designs with a compact rest state. In the positive and negative states, we expose the kirigami to magnetic fields with the same magnitude but aligned along the positive and negative y-axis directions, respectively. The design target is to achieve a given shape in the zero state and morph into different compact states given negative and positive stimuli, thus exhibiting two-way contractible behaviors. Although both the negative and positive states are compact, their panel orientations differ significantly. This characteristic holds potential for applications in photonics and phononics, as it allows for the realization of distinct chirality by manipulating the orientations of the panels[50–52].

Unlike previous cases where a single energy minimum was sought, we now aim to create two respective energy minima for the two stimuli, corresponding to the two compact states. This necessitates an extension of our method, in which the energy gradients of both compact states are aggregated into a single design objective function to identify a shared deployed design (Supplementary Note S6.2). An additional geometrical constraint is added to ensure the two compact states are both kinematically feasible (Supplementary Fig. S1f). For a more comprehensive explanation of the method, we direct readers to Supplementary Note S6.2 for detailed descriptions. We apply the extended method to achieve two-way contractible designs with circular and goblet-shape zero states, respectively (Fig. 6). The magnitude of the magnetic field in positive and negative states is maintained at $B = 35$mT, while keeping the materials and hinge parameters consistent with previous cases. By relaxing the constraint on compact shapes, we obtain a larger design space to achieve complex two-way contractible designs.

The results depicted in Fig. 6a, d demonstrate the successful attainment of the target shapes in the zero states, which transform into different compact states upon exposure to magnetic fields in opposite directions (Supplementary Movie S5 and S7). These two-way contractible behaviors align well with real physical experiments shown in Figs. 6b and 6e, and Supplementary Movie S6 and S8. In the circular design, all the magnetization orientations are almost perpendicular to

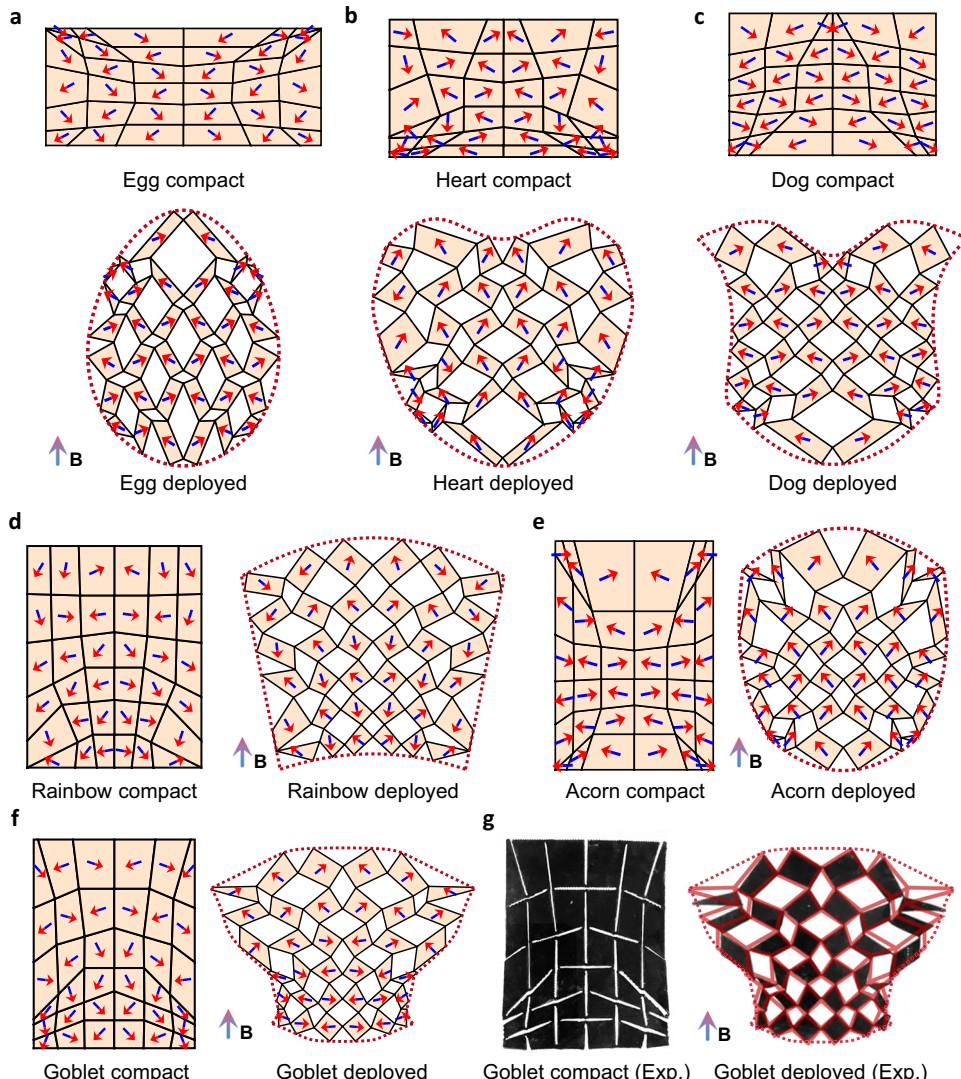

**Fig. 5 | Kirigami designs to achieve different target deployed shapes.** Compact and deployed states of optimized kirigami with (**a**) an egg-like deployed shape, (**b**) a heart-like deployed shape, (**c**) a dog-like deployed shape, (**d**) a rainbow-like deployed shape, (**e**) an acorn-like deployed shape, and (**f**) a goblet-like deployed shape. **g** Experimental verification corresponding to the optimized kirigami design in Fig. 5f, with transparent red lines marking the configuration of the simulated results for comparison. All deployed states in this figure are obtained under a magnetic field along positive vertical directions with $B = 35$ mT. All deployed shapes in this figure are rescaled for layout purposes. Unscaled results are shown in Supplementary Movie S3–S4.

the external magnetic field in the zero state, resulting in a nearly zero magnetic potential (Fig. 6c). Additionally, the cutting is designed in such a way that the rotation angle of each panel is almost identical in both compact states but with opposite signs. Consequently, the energy curves (Fig. 6c) exhibit approximate symmetry with respect to the deployed angle of the zero state, with the left and right halves corresponding to constant negative and positive fields respectively. It forms two distinct energy-decreasing paths leading to the energy minima corresponding to the two compact states. In contrast, the energy curves for the goblet-shaped design display evident asymmetry (Fig. 6f), with a larger energy change during the transition from the zero state to the negative state compared to the transition from the zero state to the positive state. As a result, both compact states can achieve physical equilibrium with the lowest total energy, but under different constant stimuli. It should be noted that there is a sudden change in the total energy in both Figs. 6c and 6f in the zero state, which is due to the change of sign for the magnetic potentials when switching the direction of external fields.

With this two-way contractible design, we have the ability to control the temporal series of external stimuli in freely transferring

between different states and realizing a desired sequence of shapes as shown in Fig. 7a, b, Supplementary Movie S5–S8. Specifically, in Fig. 7a, we start from a zero state without actuation, then impose a negative field of 35 mT to reach the negative state, and finally switch the actuation direction to get a positive field of 35 mT to realize the positive state. It is interesting to note that the kirigami exhibits an asynchronous morphing process from the negative to the positive state, i.e., starting from the left and then propagating to the right. This might be due to asymmetric manufacturing errors and friction forces. Similarly, in Fig. 7b, we can transform the kirigami to achieve positive, zero, and negative states in sequence by sequentially imposing a positive field of 35 mT, a zero field, and then a negative field of 35 mT. This capability opens up possibilities for various applications, such as wave-guiding control, locomotion in soft robotics, and mechanical computing.

## Discussion
In this study, we have proposed a differentiable design method for magneto-responsive kirigami in achieving shape morphing. Unlike existing methods that focus solely on kinematics in design followed by post-analysis, the proposed method explicitly integrates physics into

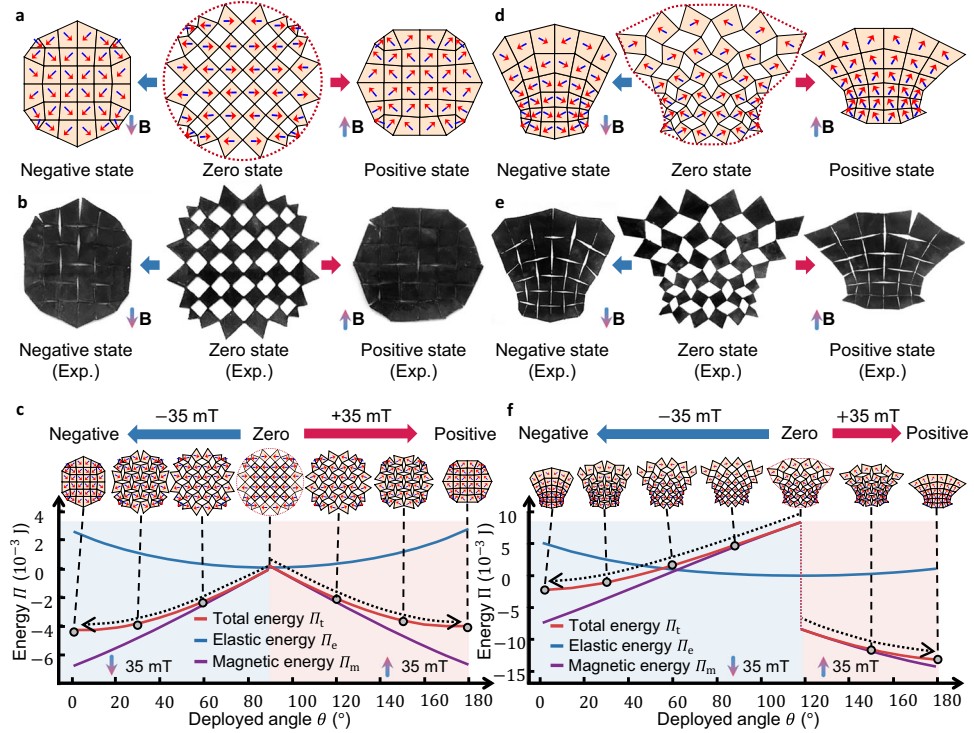

**Fig. 6 | Two-way contractible designs with target deployed shapes. a** Shows simulated configurations of designed kirigami with circular deployed shapes under negative, zero, and positive external magnetic excitations, respectively. **b** Shows experimental results corresponding to Fig. 6a. **c** Energy analysis for the kirigami in Fig. 6a, b in the constant magnetic field. Regions shaded in red and blue correspond to analysis under constant positive (+35mT) and constant negative (−35mT) magnetic excitations, respectively. Insets show kirigami configurations corresponding to the dots in the energy curves. Configurations on top of each half share the same magnetic field in energy calculation. The dashed arrows represent two contraction paths, pointing from zero states to the two energy minima (compact states). Source data are provided as a Source Data file. **d** shows different states with goblet-like deployed shapes. **e** shows experimental results corresponding to Fig. d. **f** Energy analysis for the kirigami in Fig. d, e, given the constant magnetic field. Regions shaded in red and blue correspond to analysis under constant positive (+35mT) and constant negative (−35mT) magnetic excitations, respectively. Source data are provided as a Source Data file.

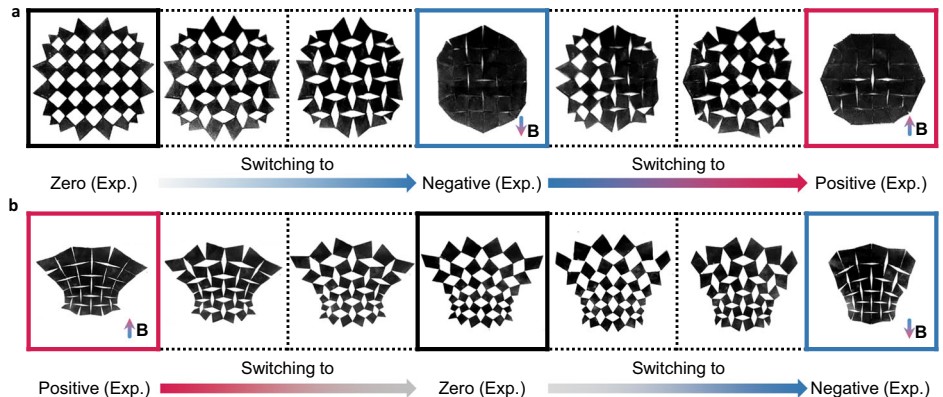

**Fig. 7 | Sequential state changes of two-way contractible designs. a, b** Show a sequence of states achieved by controlling the temporal series of stimuli for two-way contractible designs with circular and goblet-like deployed shapes, respectively.

the design loop to ensure both the kinematic and physical feasibility of the morphing process. It is built on newly developed sequential kinematic analysis and analytical energy models to capture the coupling between active materials, geometry, and stimuli. Leveraging the differentiability of our models, we formulate the design as a constrained optimization problem, simultaneously optimizing both cutting and magnetization orientations using gradient-based methods. By integrating physics into the design loop, we successfully obtain active kirigami that can be remotely actuated to morph into a wide range of complex target shapes and allow free transition between multiple

stable states via two-way contractible designs, validated through both simulations and physical experiments. It significantly reduces the computational cost of the design process (from days or weeks to minutes), demonstrating superior effectiveness and flexibility in responding to new design scenarios. Nevertheless, we acknowledge that the current design may have challenges when handling extreme shape changes or compact states with different topologies (such as prescribed holes). To address the former challenge, future research endeavors can explore more intricate kirigami unit-cell prototypes, explore the synergy of kirigami and origami techniques, or consider

panel deformation to improve flexibility. Regarding the latter challenge, there is promise in enhancing our method with the integration of a recently proposed geometric theorem[28] concerning the inverse design of kirigami tessellations with varying topologies.

Our findings shed light on the crucial role of the interplay between active materials, geometry, and stimuli in achieving physically stable morphologies. It bridges the gap between geometry and physics in active system designs, paving the way for innovative applications that require remotely actuated reconfigurability, especially in flexible electronics, minimally invasive medical treatments, and optical manipulation. For instance, it can be employed in the design of soft robots to detect and gather samples within confined spaces that are inaccessible without reconfigurability. The designed kirigami potentially can also be remotely actuated within the human body to conform to the specific shape customized to individual patients, enabling precise medical surgery or drug delivery. Moreover, hinged on the general energy principles underlying various physics, the analytical models and energy-based optimization framework in this study are applicable for stimuli other than the magnetic field, and active systems beyond kirigami, such as origami[53], lattice[54,55], tensegrity systems[56], and magnetic soft continuum robots[57].

## Methods

### Ink fabrication and preparation for direct ink writing (DIW)
The magnetic kirigami patterns are fabricated through the DIW method with ink composed of silicone-based resin with 5 μm neodymium-iron-boron (NdFeB) particles (Magnequench Co., Ltd) embedded. The ink is fabricated as follows. First, SE1700 base (Dow Corning Corp.) and Ecoflex 00-30 Part B (Smooth-on Inc.) with a volume ratio of 1:2 are mixed together at 2000 rpm for 1 min using a centrifugal mixer (AR-100, Thinky Inc.). Then, NdFeB (77.5 vol% to SE1700 base) is added to the mixture and mixed at 2000 rpm for 2 min and defoamed at 2200 rpm for 3 min. Next, SE 1700 curing agent (10 vol% to SE1700 base) is added and mixed at 2000 rpm for 1 min. The ink is transferred to a 10 mL syringe (Nordson EFD) and defoamed at 2200 rpm for 3 min and subsequently mixed at 2000 rpm for 2 min. The ink is then magnetized by a homemade magnetizer under a 1.5 T impulse magnetic field. The syringe is mounted to a customized gantry 3D printer (Aerotech) and a printing nozzle of 410 μm is utilized (Fig. S9a). CADFusion (Aerotech) is used to convert magnetic kirigami pattern drawings to G-codes for printing (Fig. S9b). The printing speed is set to 5 mm·s$^{-1}$ with extrusion pressure being 200 kPa. The printed patterns are cured at 80 °C for 36 h. More details on the printing process are included in Supplementary Note S7.

### Experiment setup for magnetic actuation
The magnetic kirigami patterns are actuated under a 1D magnetic field generated by a set of single-axis Helmholtz coils. To prevent out-of-plane deformation, magnetic kirigami patterns are covered by a supported acrylic plate.

### Simulation and optimization
Both energy-based simulation and iterative kirigami optimization are realized by sequential quadratic programming method, via the built-in function (*fmincon*) of commercial software MATLAB R2022b (The MathWorks, Inc). The commercial software Abaqus 2022 (Dassault Systèmes) is used for the finite element analysis of the kirigami to validate the proposed energy-based simulation method.

More details about manufacturing, simulation, and optimization are provided in the Supplementary Information.

### Reporting summary
Further information on research design is available in the Nature Portfolio Reporting Summary linked to this article.

## Data availability
Source data for the plots are provided with this paper. The authors declare that the data supporting the findings of this study are available within the article and its Supplemental Information files. Source data are provided with this paper.

## Code availability
The codes that support the findings of this study are available from the corresponding author upon request.

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

## Acknowledgements

We are grateful for the support from the National Science Foundation (NSF) BRITE Fellow program (Grant No. CMMI 2227641). Ruike Renee Zhao, Yilong Chang, and Shuai Wu acknowledge the support of the NSF Career Award (CMMI-2145601) and Air Force Office of Scientific Research (AFOSR YIP, FA9550-23-1-0262).

## Author contributions

L.W., Y.C., S.W., R.R.Z., and W.C. designed research; L.W., Y.C., and S.W. performed research; L.W., Y.C., S.W., R.R.Z., and W.C. analyzed data and wrote the paper.

## Competing interests

The authors declare no competing interest.
