## [Peer Review File · Nature Communications]

REVIEWER COMMENTS

Reviewer #1 (Remarks to the Author):

The authors propose a design method based on differentiable kinematics and energy models to achieve targeted shape morphing in magneto-responsive kirigami. Due to the differentiability of the developed models, the design process can be formulated as a gradient-based constrained optimization problem, therefore obviating the need to employ computationally heavy heuristic or population-based optimization algorithms previously utilized in the literature. As a result, the design process is computationally efficient. Furthermore, minimizing the total energy, consisting of elastic and magnetic portions, the proposed method takes into account both kinematics and underlying physics ensuring feasible designs from both perspectives. The outcome is optimally designed cut patterns and magnetization orientations of panels given a desired shape upon actuation and a specified magnitude of the actuating magnetic field. The paper is well written and the overall flow is good; however, there are some minor issues to be addressed:

1- In line 41 of the manuscript, “hand-pick” should be changed to “hand-picked”.

2- In the caption for Figure 1, the letters referring to the subfigures do not appear in a correct order. Apparently, it is due to the overall layout of the figure, but there should be a workaround to keep the letters in order.

3- In the caption for Figure 3, the subfigure e is referred to as “Fig. 3e”. Please consider revising the overall layout of the figure.

4- In the caption for Figure 5, several repeated sentences can be summarized into a shorter form.

5- It is mentioned in the manuscript that DIW method with a magnetized ink has been used to 3D print the designed kirigami patterns. However, it is not clearly explained how magnetization orientation in each panel is maintained. A combination of an electromagnetic coil at the location of the printing nozzle and the printing direction might have been used to align the magnetic particles according to the desired magnetization direction. As magnetization orientations are among the design variables, it is recommended to provide detailed information on how to print panels with specific magnetization orientations either in Methods section or in the Supplementary.

6- Is there a possibility of magnetic interaction between the magnetized panels? If yes, please elaborate how such interaction would affect the final configuration of the developed smart Kirigami.

7- The developed kirigami contains thin plate that is prone to out-of-plane deformation during the actuation. However, the presented results mainly show in-plane reconfiguration without considerable out-of-plane deformation caused by local instability neither in panels or hinges. The authors should elaborate how they have designed and/or analyzed the dimensions of the kirigami cuts and panels to restrain their reconfiguration to a two-dimensional one.

Reviewer #2 (Remarks to the Author):

This paper develops a method to design kirigami structures to reach target deployed states under prescribed magnetic field. The authors consider the force balance between bending of connected hinges and movement of rigid patches triggered by the magnetic field. On this basis, they set up analytical expressions about the total energies, and develop a differentiable kinematic model to evaluate the derivatives of response functions and use SQP to solve inverse optimizations. Finally, the authors verify the effectiveness of the method by designing kirigami structures with various target shapes. Overall speaking, the developed optimization method is smart and useful for designing functional kirigami structures, but its uniqueness and benefits for creating those kirigami structures that other methods could fail are not convincing. My questions are below.

1. The authors should present at least one potential real application in which the designed kirigami structures by using their method are powerful but those designed by other existing methods could fail. The authors should explain the basic idea and show the functional testing.

2. Some recent papers have already focused on design of kirigami structures with either target deformed shapes or desired functionalities. The numerical optimizations are formulated in inverse forms, and the optimization processes fully account for the equilibrium states. The authors should give a fair discussion about these works in Introduction. Some papers are

(1) Kirigami pattern design of mechanically driven formation of complex 3D structures through topology optimization, *Extreme Mechanics Letters*, 2017, 15:139-144

(2) Auxetic Kirigami Metamaterials upon Large Stretching, *ACS Applied Materials and Interfaces*, 2023, 15(15):19190-19198

3. In Figure 1, I believe the state (b) could be in equilibrium when deployed from the initial state (a). The physics equilibrium is destroyed only when mapping the deployed state (b) to state (f), since the structural outlines are transformed. Then, why the 'physics' arrow between (a) and (b) is not satisfied? Can the authors explained more clearly?

4. I do not think 'any target deployed shapes' (in row 122) can be achieved by using the developed method. My reason is that the deployed state is realized only by counter-rotation of the hinges between each two rigid patches, but no tension/compression deformations of the elastic hinges are introduced (no such energies are incorporated). It implies that the method may fail if the target shapes are to enlarge or contract the domain. The authors should discuss the feasible target shapes that the method can realize. Furthermore, the authors should discuss if the tension/compression deformations can be considered to further release the achievable deployed space.

5. Many optimized kirigami structures can be deployed to the states that 'look' like the target shapes, even though they are not 'optimal'. An example is referred to the one in the minimum energy state with the angle of 87 degrees in Figure 2, which is quite similar to the one with the angle of 100 degrees. In this sense, one may not have to develop such an objective function and to solve an optimization problem in Eq. (4). Can the authors present quantitative measure about the errors among the non-optimal (but optimized) states, the optimal state and the target shapes, and make a critical comparison?

6. Is it necessary to manufacture such simple 2D kirigami structures by using advanced 3D printing techniques?

7. I am wondering if the proposed design method can generate kirigami structures to undergo two successive but different deployed states under one applied magnetic field with different magnitudes.

8. There are many typo-errors in the manuscript; Figure 1 is easy to cause misunderstanding, as the shaded arrow looks like a design process with updated states; Figure 3 is hard to follow, as the readers may think the nodes are moved in different paths to evaluate the responses; the labels in subplots in figures are hard to follow when reading the main texts. The authors should revise the figure and the carefully examine the texts.

Response to the Reviewers' report for "Physics-aware differentiable design of magnetically actuated kirigami for shape morphing"

We would like to thank the reviewers for taking the time to thoroughly read our manuscript and provide detailed comments. The comments are insightful, which helped us improve the paper's quality. Below, we have carefully addressed the comments point-by-point. We use a separate citation list in the response letter and cite references in numerical order as they appear in the text, with the label 'R' appearing before the numerical superscript. All the changes are highlighted in yellow in the updated manuscript. Major changes are also noted with red font in this response letter.

Responses to Reviewer #1

Reviewer #1 General Comment: The authors propose a design method based on differentiable kinematics and energy models to achieve targeted shape morphing in magneto-responsive kirigami. Due to the differentiability of the developed models, the design process can be formulated as a gradient-based constrained optimization problem, therefore obviating the need to employ computationally heavy heuristic or population-based optimization algorithms previously utilized in the literature. As a result, the design process is computationally efficient. Furthermore, minimizing the total energy, consisting of elastic and magnetic portions, the proposed method takes into account both kinematics and underlying physics ensuring feasible designs from both perspectives. The outcome is optimally designed cut patterns and magnetization orientations of panels given a desired shape upon actuation and a specified magnitude of the actuating magnetic field. The paper is well written and the overall flow is good; however, there are some minor issues to be addressed..

Response: We do appreciate the reviewer's positive feedback and constructive comments. Following, we have provided point-by-point responses regarding the specific comments from the reviewer.

Reviewer #1 Comment #1: In line 41 of the manuscript, "hand-pick" should be changed to "hand-picked"

Response: We would like to thank the reviewer for catching this typo. We have corrected this in Line 39 as:

"Despite its great potential, kirigami has been largely focusing on regular periodic or heuristic cuttings, where cutting patterns are typically predetermined with hand-picked parameters rather than pursuing on-demand inverse design approaches."

and carefully checked the manuscript and SI to ensure that there are no similar mistakes.

Reviewer #1 Comment #2: In the caption for Figure 1, the letters referring to the subfigures do not appear in a correct order. Apparently, it is due to the overall layout of the figure, but there should be a workaround to keep the letters in order.

Response: We thank the reviewer for the suggestion about Figure 1. We have revised the layout, labeling, and caption of Figure 1 to make sure the subfigures appear in the correct and easier-to-read order:

Fig. 1: Schematic diagram of the physics-aware differentiable design of kirigami.

a Compact state (left) of a regular quadrilateral kirigami and its deployed state (right) transformed from the regular compact kirigami following a geometrically feasible path. The repeated four-panel unit cell is shaded in blue, where the arrow shows the magnetization orientation of each panel. **b** Geometrical compatibility requirements for a four-panel cell in edges (equal-length pairs connected by red curved arrows) and angles around the center node (red sectors). **c** Torque induced by the applied magnetic field (blue curved arrow) and the mechanical forces/torques induced by deformation (red curved arrows) are required to be in equilibrium for a physically feasible deployed state. **d** The target deployed state (right) is conformally mapped from the deployed kirigami in Fig. 1a to achieve predetermined shapes (red dashed circular contour as an example) under the magnetic field B . However, there is no compact state (left) that can follow a geometrically and physically feasible path to this mapped deployed state. **e** Compact state (left) and deployed state (right) of the kirigami optimized from the conformally mapped design in Fig. 1d, satisfying both physical equilibrium and geometrical compatibility. **f** This physics-aware optimization integrates two models, the differentiable kinematic model (top) and the differentiable energy model (bottom) under a fixed external magnetic field. The enlarged inset shows the

equivalent nonlinear springs for kirigami hinges. The optimization process achieves physical equilibrium by ensuring minimal energy of the deployed kirigami indicated by zero gradients.

Reviewer #1 Comment #3: In the caption for Figure 3, the subfigure e is referred to as “Fig. 3e”. Please consider revising the overall layout of the figure.

Response: We have revised Figure 3 and its caption to improve its readability as follows:

Fig. 3: Differentiable kinematic model.

a Two exemplified panels in kinematic analysis corresponding to the red dashed box in the subfigure below. The grey dot is fixed and the locations of the remaining colored nodes are solved, which can be divided into three basic geometry problems indicated by shades in different colors. **b** Details of three basic geometry problems to be solved sequentially. In each problem, nodes in colors can be analytically solved based on the fixed nodes (grey) and already-solved nodes (white). The solution of each problem can be considered as a transformation from known nodes to nodes to be solved. **c** Extracted forward computational graph (left) from Fig. 3b by composing the transformation in sequence. Different types of transformations are marked by arrows in different colors, pointing from preceding nodes to the updated nodes. Shaded regions represent geometry problems, from which the analytical expression of the transformations f_i are obtained. A simplified graph is shown on the right. X_0 and θ are the original node locations and deployed angle, respectively. X_i is the set of updated nodes obtained from each transformation,

corresponding to nodes of the same color in the original graph. **d** Backward computational graph to calculate the gradient of locations for the node of interest by composing the gradient of individual transformation ∂f_i via chain rules. **e** Sequential solving process to obtain all nodal locations. Only the right half is shown here as the left half is obtained by mirror symmetry on the center line. **f** Forward computational graph corresponding to sequential steps in Fig. 3e. The analytical transformations in each step are marked by arrows in different colors. Nodes that have been updated in previous steps are colored in white. **g** Backward computational graph (marked in red) to calculate the gradient of a given node.

Discussion related to Figure 3 is also updated in line 247, page 12 of the revised manuscript: “Given any change in the deployed angle, we solve for the corresponding kirigami configuration (kinematic analysis) sequentially as shown in Fig. 3. Specifically, calculating nodal locations of kirigami panels (Fig. 3a) involves a series of simple geometrical problems (marked by shaded regions in Fig. 3b), in which we solve for the unknown nodes (colored nodes) based on their constraints with the nodes obtained in preceding problems. Solutions for nodal locations in these basic problems can be analytically derived, which can be viewed as a series of analytical transformations f_i (marked by colored arrows in Fig. 3c) from the preceding nodes to unsolved nodes (X_0 to X_3 and so on). By composing these transformations in sequence, they form a directed computational graph for forward analysis (Fig. 3c). Using chain rules, we can obtain the gradient values of node locations of any nodes by tracing the computational graph backward and multiplying the gradient of basic transformation ∂f_i in order (Fig. 3d). We apply this kinematic analysis to the whole kirigami, iterating nodes in every column of panels or voids (Fig. 3e) and sequentially composing the transformations together to get the analytical expression for the kirigami configuration. This analytical expression can be represented by a directed computational graph in Fig. 3f. Note that, due to the inherent regularity of cutting topology, i.e., the interconnections between panels, the solutions in each column only involve a limited set of transformation types (indicated by arrows in different colors in Fig. 3f). Using the chain rule again, we can readily calculate the analytical gradient of any nodal locations with respect to the deployed angle θ and initial node locations X_0 (Fig. 3g).”

Reviewer #1 Comment #4: In the caption for Figure 5, several repeated sentences can be summarized into a shorter form.

Response: Thank the reviewer for the suggestion to improve our manuscript's readability. We have revised the Fig. 5 caption and combined the labeling of compact and deployed shapes together for simplicity:

Fig. 5: Kirigami designs to achieve different target deployed shapes.

a-f Compact and deployed states of optimized kirigami with **(a)** an egg-like deployed shape, **(b)** a heart-like deployed shape, **(c)** a dog-like deployed shape, **(d)** a rainbow-like deployed shape, **(e)** an acorn-like deployed shape, and **(f)** a goblet-like deployed shape. **g** Experimental verification corresponding to the optimized kirigami design in Fig. 5f, with transparent red lines marking the configuration of the simulated results for comparison. All deployed states in this figure are obtained under a magnetic field along positive vertical directions with $B = 35$ mT. All deployed shapes in this figure are rescaled for layout purposes. Unscaled results are shown in Supplementary Video S3-S4.

Reviewer #1 Comment #5: It is mentioned in the manuscript that DIW method with a magnetized ink has been used to 3D print the designed kirigami patterns. However, it is not clearly explained how magnetization orientation in each panel is maintained. A combination of an electromagnetic coil at the location of the printing nozzle and the printing direction might have been used to align the magnetic particles according to the desired magnetization direction. As magnetization

orientations are among the design variables, it is recommended to provide detailed information on how to print panels with specific magnetization orientations either in Methods section or in the Supplementary.

Response: We thank the reviewer for the suggestion and have added a new Supplementary Note S7 to provide more details about the DIW printing of the magnetic kirigami:

“The magnetic kirigami patterns are created using the Direct Ink Writing (DIW) method, employing ink prepared according to the procedure outlined in the Methods section. The ink is magnetized using a homemade magnetizer under a 1.5 T impulse magnetic field. The syringe is attached to a customized gantry 3D printer (Aerotech) featuring a 410 μm printing nozzle for the fabrication process. With an attached permanent magnet on the nozzle, the polarities of magnetized particles in the ink are aligned to the printed direction during extrusion (Fig. S9a). The kirigami panel magnetization is thus programmed by controlling the filament printing direction. Note that there is a magnetic field shield between the permanent magnet and the printing platform so that the magnetizations of printed filament remain unchanged when the printing nozzle is moving around. For a given optimized kirigami design (Fig. S9b), a printing pattern drawing is created with the encoded printing direction of each panel while considering the kirigami hinge dimensions. Then, CADFusion (Aerotech) is used to convert magnetic kirigami pattern drawings to G-codes for printing. The printing speed is set to 5 $\text{mm}\cdot\text{s}^{-1}$ with extrusion pressure being 200 kPa. The printed patterns are cured at 80°C for 36 h.”

Fig. S9. Direct ink writing printing process of magnetic kirigami. **a** Schematic direct ink writing printing of magnetic composite for a panel with programmed magnetization. **b** The optimized kirigami design, the converted printing pattern, and the printed sample with the circular deployed state in Fig. 4a~d.

Reviewer #1 Comment #6: Is there a possibility of magnetic interaction between the magnetized panels? If yes, please elaborate how such interaction would affect the final configuration of the developed smart Kirigami.

Response: We thank the reviewer for bringing up this interesting question. There is magnetic interaction between the panels if they are close to each other. However, the surface magnetic field generated by the printed kirigami panel is measured to be very small, in the one millitesla to sub-millitesla range, and it decays rapidly with increasing distance from the panel. Thus, these interactions between panels are weak and have a negligible contribution to the actuation of the kirigami under a magnetic field. This assertion is further substantiated by that fabricated samples

retain the as-printed shapes without seeing obvious attraction/repulsion between panels. We have incorporated a clarifying note in Line 232 of the manuscript to emphasize this point:

“Note that, the magnetic interaction between panels is very small and it has a negligible effect on the magnetic actuation of the printed kirigami, which is thus ignored here.”

Reviewer #1 Comment #7: The developed kirigami contains thin plate that is prone to out-of-plane deformation during the actuation. However, the presented results mainly show in-plane reconfiguration without considerable out-of-plane deformation caused by local instability neither in panels or hinges. The authors should elaborate how they have designed and/or analyzed the dimensions of the kirigami cuts and panels to restrain their reconfiguration to a two-dimensional one.

Response: We thank the reviewer for raising this question about how we handle the potential out-of-plane deformation. The primary focus of our research is 2D kirigami. The goal is to demonstrate that even when we explicitly account for multiple underlying physical phenomena during actuation, we can still achieve high efficiency and differentiation in the design process by carefully defining the optimization problem. Consequently, we have assumed that the shape-morphing behaviors remain strictly in-plane, without any out-of-plane deformations, aligning with the central theme of our investigation.

Furthermore, it is worth noting that, the kirigami system in our study differs from classical thin plate models in a fundamental way. In classical thin plates, out-of-plane buckling typically occurs under compressive loads^{R1}. In contrast, our magnetic fields induce in-plane torsional forces in each panel. Since the rotation of the panel is kinematically compatible with its neighboring panels, the primary response of panels in our system is in-plane rigid rotation rather than compressive deformation. Meanwhile, the hinges predominantly experience in-plane bending, and they possess a relatively small aspect ratio $l/h \approx 1.33$. As a result, the kirigami system is inherently less prone to buckling compared to classical thin plates, highlighting one of the advantages of kirigami over continuous materials^{R2}.

Therefore, in the design process, we assume the kirigami has enough thickness to resist buckling. This assumption can be easily incorporated into the design process because, as we emphasized in the SI, both types of energy involved in the system are linearly related to thickness, ensuring that their equilibrium remains independent of thickness variations. In other words, changes in thickness do not influence the design outcomes or the physical equilibrium of the system.

In Line 306, we added a sentence to give more explanation on this:

“It should be noted that, since both types of energies are linearly related to the thickness of the kirigami, the thickness value will not influence the design outcomes or the physical equilibrium of the system. Therefore, during the design process, we assume the thickness to be large enough in real applications to keep the kirigami always in-plane.”

Furthermore, in our experiments, we take extra measures to prevent out-of-plane deformation. Specifically, we cover the magnetic kirigami patterns with a supported acrylic plate, further reinforcing the confinement of the deformation within the plane of interest.

Responses to Reviewer #2

Reviewer #2 General Comment: This paper develops a method to design kirigami structures to reach target deployed states under prescribed magnetic field. The authors consider the force balance between bending of connected hinges and movement of rigid patches triggered by the magnetic field. On this basis, they set up analytical expressions about the total energies, and develop a differentiable kinematic model to evaluate the derivatives of response functions and use SQP to solve inverse optimizations. Finally, the authors verify the effectiveness of the method by designing kirigami structures with various target shapes. Overall speaking, the developed optimization method is smart and useful for designing functional kirigami structures, but its uniqueness and benefits for creating those kirigami structures that other methods could fail are not convincing. My questions are below.

Response: We appreciate the reviewer's efforts in reviewing our manuscript. In the revised manuscript, we have taken great care to emphasize the distinctiveness and advantages of our proposed method. Before diving into the detailed point-to-point response to the reviewer's specific comments, we would like to provide a thorough discussion concerning the strengths and uniqueness of our approach.

Our study primarily centers on the advancement of design methodologies specifically tailored for active shape-morphing kirigami. In this context, we offer a comprehensive overview of the current state-of-the-art from a design perspective, highlighting key areas where existing approaches may fall short in meeting the demands of active shape morphing. In this overview, we use "SOTA" for the features of state-of-the-art and "Need" for the requirements of kirigami driven by stimuli-responsive materials:

1) Design representation:

a. Unit-cell design vs. Aperiodic system design

SOTA: Most existing designs focus on a single unit-cell kirigami, its periodic assembly, or heuristic cuttings.

Need: Realizing complex deployed shapes necessitates the use of aperiodic kirigami systems rather than periodic unit-cell designs^{R3}. The rationale behind this lies in the fact that periodic designs result in uniform local area changes throughout the entire system, limiting the shape-morphing capability.

b. Geometrical parameters vs. Material parameters

SOTA: The design space typically only involves geometrical design parameters, often neglecting material and stimuli-related factors.

Need: For active kirigami driven by stimuli-responsive materials, we need the co-design of both geometry and material parameters (such as the magnetization orientation of the magnetic origami in this work), extending beyond purely geometrical considerations.

c. Dimension of design space and complexity of constraints

SOTA: The design space is usually low-dimensional with a limited number of constraints.

Need: Aperiodic design and material parameters inherently result in a high-dimensional design space with numerous nonlinear constraints.

2) Design evaluation:

a. Mechanical loading vs. Multiple physics

SOTA: The majority of research evaluates the design only by geometrical compatibility and forward kinematic analysis. Some research uses finite element analysis (FEA) to consider physics, but these are generally limited to passive kirigami subjected solely to mechanical loading.

Need: Simulating active kirigami involves a delicate balance between mechanical and external stimuli, distinguishing it from passive kirigami subjected only to mechanical loading.

b. Efficiency and differentiability

SOTA: Existing methods rely on standard FEA, which is relatively time-consuming and not differentiable.

Need: Given the high-dimensional nature of the design problem and the presence of many nonlinear constraints, it is highly desirable to have an efficient but accurate evaluation method, with analytical gradients for objectives and constraints.

3) Design synthesis:

SOTA:

- Many kirigami research assumes predetermined cutting patterns with hand-picked parameters, or exhaust the whole low-dimensional design space, rather than pursuing on-demand inverse design approaches.
- Inverse design approaches for kirigami often ignore the physics.
- Physics-aware designs typically rely on computationally intensive FEA for evaluation, lacking analytical gradients and consequently resorting to gradient-free heuristic optimization methods

Need: Physics-aware optimization methods are needed to address high-dimensional design problems with many nonlinear constraints, without requiring an extensive number of evaluations as seen in gradient-free heuristic optimizers.

Upon contrasting the capabilities of the current state-of-the-art with the specific needs of active shape-morphing design, it becomes apparent that they do not align seamlessly. Our method exhibits its strengths and uniqueness, because it aims to advance design methods for active shape-morphing kirigami and can effectively meet the aforementioned needs in different aspects. In our response to Reviewer #2 Comment #1, we also provide a quantitative comparative study to provide further support.

We have made substantial revisions to the Introduction to provide a critical assessment of existing methods and to better elucidate our motivations. At the beginning of the second paragraph, we added more detailed discussions on methods that do not consider physics:

“Despite its great potential, kirigami has been largely focusing on regular periodic or heuristic cuttings, where cutting patterns are typically predetermined with hand-picked parameters rather than pursuing on-demand inverse design approaches¹³. Several studies have explored inverse design methods, mainly focusing on parameter optimization, such as hinge widths and panel aspect ratios^{24,25}. Moreover, these studies have primarily been conducted for the unit cell of periodic kirigami configurations, which will restrict the flexibility in realizing complex shapes. To further harness kirigami's shape-morphing capabilities for intricate functionalities, a recent inverse design framework has successfully relaxed the periodicity requirement in kirigami, creating optimized aperiodic cutting patterns to realize versatile deployed shapes²⁶.”

At the end of the second paragraph, we elaborate on why several physics-aware exceptions do not meet the requirement of this study:

“There are a few physics-aware exceptions, most of which only consider physics in post-analysis after completing the design process^{26,27,32} or assembling precomputed unit cells with weak interactions³³⁻³⁷. In instances where physical simulations are directly integrated into the inverse design loop, they often incorporate full finite element analysis (FEA)^{16,19,20,22,23,38,39} and primarily account for mechanical loading only. Moreover, these design processes frequently resort to brute-force parameter sweeping or gradient-free heuristic optimizers. The applicability of these optimizers, combined with the resource-intensive nested FEA, confines the scope of these designs to low-dimensional problems with simple constraints, such as periodic or unit-cell designs. Overall, there is a lack of an efficient and flexible physics-aware inverse design framework, which hampers the integration of kirigami with stimuli-responsive materials to realize more complex and practical applications.”

In the consecutive paragraph, we define the main challenges to be addressed in this study:

“The key barrier in designing physics-aware active kirigami is to incorporate the complex interplay between geometry, materials, and external stimuli in an iterative, automated design process. Furthermore, simulating the deployment process is often time-consuming and non-differentiable, without the analytical gradient of the design objective required to effectively navigate in a high-dimensional design space. As a result, the consideration of multiple physics often appears incompatible with high design flexibility and efficiency. This study addresses these issues and demonstrates how to explicitly and yet efficiently incorporate both geometry and physics into kirigami design for active shape morphing via differentiable modeling and gradient-based optimization.”

Reviewer #2 Comment #1: The authors should present at least one potential real application in which the designed kirigami structures by using their method are powerful but those designed by other existing methods could fail. The authors should explain the basic idea and show the functional testing.

Response: We recognize that the reviewer's inquiry regarding real-world applications implicitly touches upon the advantages and distinctiveness of our method compared to other existing kirigami designs, which we have elaborated in our proceeding response to the reviewer's general comment. The primary focus of our current study is on the development of general methods from the design perspective. Like many existing works on programmable kirigami design methods, due to the feasibility of fabrication, our paper is focused on presenting our new approach and its uniqueness using an example that we can fabricate and physically validate. To address the reviewer's concern,

we added a description of the potential real application of our example problem, providing more contextual information for the readers. We also provide a comparative analysis of our method versus others with quantitative measures.

1. Potential Applications: We have included a remark at the end of the Conclusion to address this aspect:

“It bridges the gap between geometry and physics in active system designs, paving the way for innovative applications that require remotely actuated reconfigurability, especially in flexible electronics, minimally invasive medical treatments, and optical manipulation. For instance, it can be employed in the design of soft robots to detect and gather samples within confined spaces that are inaccessible without reconfigurability. The designed kirigami potentially can also be remotely actuated within the human body to conform to the specific shape customized to individual patients, enabling precise medical surgery or drug delivery.”

2. Comparative Performance: Can we use other design methods to achieve comparable performance in this particular scenario?

The primary performance metric for the aforementioned applications is the degree to which the deployed shape aligns with the intended design target after actuation. As elucidated in our response to the general comment, we have underscored that existing methods do not satisfy the design requirements in this particular case. Nevertheless, we have conducted a comprehensive comparative analysis to provide additional insights into the strengths and distinctive qualities of our approach concerning this specific realm of active shape morphing.

1) Key quantitative criteria for the comparison

In our comparative analysis, we assess various methods based on two key criteria: computational cost and the degree of alignment with the target shapes. To gauge the computational cost, we consider factors such as computational time and the number of objective function calls required for each method. Regarding the alignment with target shapes, we employ a quantitative measure. This measure, which we originally utilized as a constraint in the optimization process (detailed in S1.4 of the Supplementary Information), has been modified in this response letter to provide a quantitative assessment of how closely the deployed shape matches the intended design target.

Fig. R1: Quantitative measure

Specifically, as shown in Fig. R1 (also in Fig. S1d), we use a spline to represent the target shape in the deployed state (red dash line). We then measure the deviation of each boundary node V_i by the distance to its projected node \hat{V}_i on the spline, which also corresponds to the shortest distance between V_i and the target shape. We can then define the quantitative error measure of the whole kirigami σ to be the sum of the deviation of all the boundary nodes normalized by the characteristic length ℓ (the largest dimension) of the target shape:

$$\sigma = \sum_i \|\mathbf{x}_i - \hat{\mathbf{x}}_i\| / \ell, \quad (\text{R1})$$

where \mathbf{x}_i and $\hat{\mathbf{x}}_i$ are the coordinates of V_i and \hat{V}_i , respectively. Ideally, this measure should be zero. The smaller this value, the better the design.

2) Case study 1: Analytical gradient (proposed) vs. Numerical gradient

Using the defined quantitative measure, our initial comparison focuses on contrasting our differentiable design method, which employs analytical gradients, with a non-differentiable design method that relies on numerically approximated gradients. It's worth noting that both methods share identical optimization problem definitions and settings; the sole distinction lies in the manner in which gradients are computed. This is to highlight the critical role of using differentiable modeling and optimization, a fundamental contribution and unique aspect of our proposed method.

We employed both methods to design kirigami structures with goblet-like deployed shapes, as depicted in Fig. R2. It becomes evident that when using our method with analytical gradients, the deployed shape closely aligns with the target shape. Conversely, when relying on numerical gradients, a notable disparity emerges between the designed and target shapes. To provide a quantitative comparison of the design outcomes, we present the results in Table R1.

Fig. R2: Design results obtained by methods with the **a** analytical gradient, and **b** numerical gradient

Table R1. Quantitative measures for the performance of different methods

Method	# of iterations	# of function calls	Execution time	Error σ
Analytical gradient (proposed)	54	230	58.53s	2.1548×10^{-4}
Numerical gradient	146	30646	3h 55min 56.12s	0.5124
Genetic algorithm	Terminated after one generation, no initial feasible design found			

The results clearly demonstrate several advantages of our proposed method. Firstly, it achieves significantly faster convergence, requiring fewer function calls and exhibiting much higher efficiency, with a remarkable $\sim 240\times$ speedup when compared to the method employing numerical gradients. Additionally, our method yields superior shape morphing performance, as evidenced by significantly lower error values. This difference in performance between the two methods arises because the numerical gradient requires numerous additional function calls to approximate real values, and the approximation potentially leads the optimizer towards incorrect optima. This underscores the critical importance of achieving differentiability in both the modeling and optimization processes, a feature often absent in most existing methods but successfully enabled in our approach.

3) Case study 2: Gradient-based optimizer (proposed) vs. Heuristic optimizer (GA)

As previously discussed, the majority of existing physics-aware kirigami designs for various applications typically combine genetic algorithms (GA) with non-differentiable FEA. However, these approaches are primarily tailored for unit-cell or periodic designs, focusing on passive kirigami with only mechanical loading. While the enhancement of existing methods falls outside the scope of our study, we have still extended the GA-based methods to accommodate aperiodic kirigami structures for magnetically actuated shape-morphing behaviors.

In the extended approach, we employed the default GA optimizer available in MATLAB, initializing the process with a population of 100 designs. We included the mapped design depicted in Fig. 1d within the initial GA population. It's important to mention that in existing GA-based methods, a differentiable energy model, like the one used in our original optimization objective, is typically not available. Therefore, we replaced the original energy-based objective function that requires gradients with the error measure σ and removed the target shape constraint. To avoid unaffordable computational costs, we replaced the FEA with our energy-based simulation method.

We applied the revised GA-based method to design kirigami structures with the same goblet-like target shapes as shown in Fig. R2. However, this GA-based approach encountered significant challenges during its execution. As indicated in Table. R1, it terminated after only one iteration since it could not find any feasible solutions, which already took nearly four hours ($\sim 240 \times$ of our method with converged solution). The primary reason for this outcome is the complexity of the design problem, which encompasses 204 design variables, 120 nonlinear inequality constraints, and 134 nonlinear equality constraints. This high-dimensional problem, coupled with a multitude of nonlinear constraints, presents an extremely challenging scenario for constrained genetic algorithms. Furthermore, if we were to employ full FEA, as done in existing literature, the computational expense would become prohibitively higher. It's worth noting that utilizing more advanced GA or other gradient-free heuristic optimizers could potentially yield improved results in this context. Nevertheless, we believe such an approach may not be necessary, given that our method has already demonstrated superior performance with significantly reduced complexity.

We believe this comparative study provides additional evidence of the exceptional efficiency and effectiveness of our methods in addressing this specific design scenario, as a direct response to the reviewer's comment. Nonetheless, we have opted not to include these results in the manuscript as they fall outside the scope of our primary study.

Reviewer #2 Comment #2: Some recent papers have already focused on design of kirigami structures with either target deformed shapes or desired functionalities. The numerical optimizations are formulated in inverse forms, and the optimization processes fully account for the equilibrium states. The authors should give a fair discussion about these works in Introduction. Some papers are

(1) Kirigami pattern design of mechanically driven formation of complex 3D structures through topology optimization, *Extreme Mechanics Letters*, 2017, 15:139-144

(2) Auxetic Kirigami Metamaterials upon Large Stretching, *ACS Applied Materials and Interfaces*, 2023, 15(15):19190-19198

Response: We thank the reviewer for bringing these references to our attention. Both of these papers indeed integrate finite element analysis (FEA) into the design process, demonstrating the

successful inverse design of kirigami systems to attain specific functionalities. Their contributions in their respective domains are noteworthy.

On the other hand, it's important to note that the specific focus of our study pertains to the active shape morphing design, which may differ in certain aspects from the applications presented in those papers. We have provided a comprehensive comparison of the state-of-the-art (SOTA) capabilities and the specific design requirements of this study in our response to the general comment. Here, we will delve into a more detailed discussion of these two papers to further clarify the distinctions and contributions.

First, the primary focus of our study centers on the following key aspects:

- 1) Designing aperiodic kirigami systems with enhanced flexibility.
- 2) Target shape morphing as the design objective.
- 3) Designing magnetically actuated kirigami, characterized by the intricate interplay between magnetic and elastic forces.
- 4) Explicitly considering physics while maintaining differentiability to facilitate the efficient gradient-based inverse design.

For the first cited paper, it's worth noting that this work was already introduced in the original manuscript's introduction (Ref 38). Our study and this work have different focuses, diverging in each of the four aforementioned aspects:

- 1) Their focus primarily centers on unit-cell design, which is appropriate for the application in their paper. In contrast, our focus is on aperiodic kirigami systems with high-dimensional design space, and numerous complex constraints.
- 2) While their work employs surface curvature as the design target, our emphasis is on shape morphing, specifically characterized by contour changes, setting our study apart in terms of design objectives.
- 3) Their system operates as a passive kirigami system with mechanical loading. Our focus is on active shape morphing involving both magnetic and mechanical forces, which introduces an additional layer of complexity.
- 4) Physics considerations are integrated through finite element analysis (FEA) in their paper, which is not differentiable and thus provides no analytical gradient. Consequently, it utilizes heuristic genetic algorithms, i.e., GA, for optimization that does not require gradient information. While it is undoubtedly effective for the application in their paper, it might not meet the requirement of this study, which is featured by high-dimensional design space and multiple nonlinear constraints. Please refer to the second comparative study in our response to Reviewer #2 Comment #1, in which we compared our method with a GA-based method.

The second referenced paper (Ref 39 in the revised draft) has some new advancements:

- 1) It also focuses on unit-cell design in a periodic kirigami.
- 2) Its design target focuses on the Poisson's ratio, emphasizing the lateral displacement of two nodes. This differs from our focus on shape morphing, which involves spatially varying displacement requirements for all boundary nodes.

- 3) Its application focuses on passive kirigami with mechanical loading.
- 4) In their paper, physics considerations are incorporated via non-differentiable FEA, with genetic algorithms as the optimizer, followed by a fine-tuned stage using numerical gradients. While this method exhibits good performance for periodic kirigami designs, numerical gradients are costly to compute in our multi-physics aperiodic designs. This is due to the high-dimensional aperiodic design space, a large number of constraints, and multi-physics simulations. Please refer to the first comparative study in our response to Reviewer #2 Comment #1, in which we compared our method with a method using numerical gradients.

Indeed, these referenced methods have demonstrated commendable performance within their respective application. However, based on these discussions, it can be concluded that our work is fundamentally different and exhibits its strengths and uniqueness regarding the inverse design of magnetically actuated kirigami. We cited the two references and modified the second paragraph in the Introduction section. Please refer to the first part of our response to Reviewer #2 Comment #1.

Reviewer #2 Comment #3: In Figure 1, I believe the state (b) could be in equilibrium when deployed from the initial state (a). The physics equilibrium is destroyed only when mapping the deployed state (b) to state (f), since the structural outlines are transformed. Then, why the ‘physics’ arrow between (a) and (b) is not satisfied? Can the authors explained more clearly?

Response: Thank the reviewer for bringing up this question! We would like to point out that physical equilibrium in this study refers to the balance between the magnetic torque acting on the panels and the elastic forces exerted by the hinges. Therefore, to determine whether a kirigami is in physical equilibrium, one needs to carefully examine whether the magnetic forces are in equilibrium with the elastic forces, or whether the total energy of the system reaches its minimum.

In the case of the deployed state as depicted in the original Figure 1b (now Figure 1a, $\theta = 90^\circ$ in Fig. R3), it is evident from the energy analysis in Fig.R3 that, this state does not attain physical equilibrium under the applied magnetic field of $B = 30$ mT, even before any mapping occurs. This is because the magnetic torques generated are insufficient to counterbalance the elastic forces induced by the bending of the hinges. Instead, the equilibrium state corresponds to $\theta = 78^\circ$ where the total energy is at the local minimum. This confirms the fully deployed state is not in physical equilibrium.

In fact, the key issue is that achieving equilibrium is contingent upon the interplay between external fields and materials. However, the regular kirigami configuration depicted in Figure 1a is predefined and does not necessitate any specific information regarding the external field, materials, or hinge parameters. In other words, even if we were to alter the external field or utilize different sizes of hinges, this regular design would remain unaltered. Consequently, it becomes evident that this fixed design cannot inherently fulfill the requirement for physical equilibrium across all possible scenarios. This is the very reason we add a crossing mark to the arrow between the two states to emphasize that this transition is usually not physically feasible.

Fig. R3: Energy analysis of the kirigami in the original Fig. 1b (now in Fig. 1a) with $B=30$ mT.

As another example, in Fig. R4, we show the energy analysis given a weaker magnetic field $B=20$ mT. In this case, the kirigami exhibits identical elastic energy levels but significantly reduced magnetic potential in comparison to the case represented in Figure R3, where $B=30$ mT. Consequently, the deployed shape, as depicted in the original Figure 1b (now Figure 1a), deviates further from the state of physical equilibrium. In contrast, using our method can rapidly respond to this new scenario and retain the target deployed shape as shown in Fig. 4g.

To avoid any potential misconceptions regarding the concept of physical equilibrium, we added explanatory text to Figure 1c. Please refer to our response to Reviewer #2 Comment #8. We also modified the caption to emphasize that an equilibrium state requires the balance of both magnetic and elastic forces, instead of solely examining the magnetic forces or geometry:

“ **c** Torque induced by the applied magnetic field (blue curved arrow) and the mechanical forces/torques induced by deformation (red curved arrows) are required to be in equilibrium for a physically feasible deployed state. ”

Fig. R4: Energy analysis of the kirigami in the original Fig. 1b (now in Fig. 1a) with $B=20$ mT.

Reviewer #2 Comment #4: I do not think ‘any target deployed shapes’ (in row 122) can be achieved by using the developed method. My reason is that the deployed state is realized only by counter-rotation of the hinges between each two rigid patches, but no tension/compression deformations of the elastic hinges are introduced (no such energies are incorporated). It implies that the method may fail if the target shapes are to enlarge or contract the domain. The authors should discuss the feasible target shapes that the method can realize. Furthermore, the authors should discuss if the tension/compression deformations can be considered to further release the achievable deployed space.

Response: We thank the reviewer for raising this question about the applicability of our method. We would like to respond by answering the following three questions:

1. Can the proposed method accommodate target shapes that have the overall domain enlarged or contracted from its initial state?

While the reviewer is correct that the kirigami system in our study only involves counter-rotation of rigid panels, it does not mean that it can only achieve target deployed shapes with the same domain area as their compact states.

- 1) It can achieve deployed shapes with an enlarged domain area. In fact, all the deployed shapes shown in this study do have the overall domain enlarged compared with their compact states. To illustrate this, we use the circular designs in Fig. 4a~b as the example, and simultaneously show their compact and deployed states on the same scale below:

Fig. R5: Deployment of the circular design and the corresponding domain change

From the figure, it can be noted that the domain area in the deployed state expands by 2.1331 times compared with its compact state.

- 2) It can not achieve a deployed shape with a smaller domain area than its compact state. The reason is that, by definition, a compact state does not contain any voids between the panels. Since the panels are assumed to be rigid in our study, the area of the compact state is the lower bound of areas for any achievable shapes. However, it is arguable that labeling the target shape as a "deployed" state might be inappropriate if it is known to be smaller than the compact state. A more practical and applicable way would be to designate the state with a larger area as the deployed state, which can be effectively handled by our method. This way also aligns more closely with practical applications, as it enables the reduction of space required for storing the kirigami when it is not in active use.
 - 3) The key point we would like to clarify in addressing this question is the fundamental distinction between how kirigami systems achieve shape morphing compared to continuum material systems. As we mentioned in the introduction, in a continuum material system, there are stringent continuity requirements for the solid, necessitating very large local deformations and strains in the material to facilitate shape morphing. Conversely, kirigami relies on the kinematic movement of panels, leading to the redistribution of the solid material and the creation of voids in various regions. Therefore, it allows the significant local deformations to be accommodated within the voids rather than the solids, thereby enabling more versatile and flexible shape-morphing behaviors compared to their continuum counterparts, even when the panels are rigid.
2. What target shapes are feasible to realize with the proposed method?

While we didn't encounter significant challenges in achieving the diverse target deployed shapes shown in our current study, we acknowledge the reviewer's point that there should be some constraints on the capabilities of our method. To ensure that we do not overstate the universality of our approach, we have adjusted the sentence in Line 123 accordingly:

“Our goal is to develop a fully automated inverse design approach so that **for various given target deployed shapes**, the kirigami cutting and magnetization of each panel can be rapidly obtained to achieve the desired shapes after actuation while ensuring both geometrical and physical feasibility.”

Meanwhile, establishing rigorous proof of the comprehensive applicability of the proposed approach would require extensive investigations, if not impossible. This endeavor would entail a combination of mathematical theories to determine the limits of geometric changes and the

development of accompanying physical models to explore extreme scenarios where external energy might prove insufficient to maintain the desired deployed shape. While the pursuit of such a comprehensive theoretical proof falls beyond the primary scope of this study, we have augmented the revised manuscript with additional notes in the second to the last paragraph of the Conclusion, providing insights from our design experience:

“Nevertheless, we acknowledge that the current design may have challenges when handling extreme shape changes or compact states with different topologies (such as prescribed holes). To address the former challenge, future research endeavors can explore more intricate kirigami unit-cell prototypes, explore the synergy of kirigami and origami techniques, or consider panel deformation to improve flexibility. Regarding the latter challenge, there is promise in enhancing our method with the integration of a recently proposed geometric theorem²⁸ concerning the inverse design of kirigami tessellations with varying topologies.”

3. Can we increase the flexibility of the proposed method if tension/compression deformations are considered?

Yes, the flexibility of our proposed method could certainly increase with the inclusion of tension/compression deformations. However, the extent of improvement may not be substantial. The reason is that, as we illustrated in the first question, the shape morphing of kirigami mainly relies on the kinematic movements of panels rather than the local deformation. Also, the external magnetic field may not possess the requisite strength to achieve significant in-plane deformations of the panels.

Reviewer #2 Comment #5: Many optimized kirigami structures can be deployed to the states that 'look' like the target shapes, even though they are not 'optimal'. An example is referred to the one in the minimum energy state with the angle of 87 degrees in Figure 2, which is quite similar to the one with the angle of 100 degrees. In this sense, one may not have to develop such an objective function and to solve an optimization problem in Eq. (4). Can the authors present quantitative measure about the errors among the non-optimal (but optimized) states, the optimal state and the target shapes, and make a critical comparison?

Response: We agree with the reviewer's observation that in certain instances, the deployed shapes may appear similar to the target shape even without considering the underlying physics. However, it is essential to stress that this similarity cannot always be guaranteed, as the critical physical requirements are not factored into the design process.

Furthermore, we would like to emphasize that our method explicitly incorporates physics to achieve both physical feasibility and superior performance without significantly compromising design efficiency. Therefore, we believe that opting for our proposed method to attain the optimal design is the more prudent choice, as opposed to relying on the small chance of occurrence of a shape that merely resembles the target.

As suggested by the reviewer, we defined a quantitative measure (see Part 1 in our response to Reviewer #2 Comment #1) to compare different designs, with the values given below:

1. $\sigma_1 = 0.3289$ for the real deployed shape without physics-aware optimization (energy minimum with $\theta = 87^\circ$ in Fig. 2g);

2. $\sigma_2 = 0.0004$ and the real deployed shape with physics-aware optimization (kirigami in Fig. 4b, energy minimum in Fig. 4e);

Please note that a smaller value of σ indicates a better alignment between the deployed shape and the target shape. The results clearly demonstrate the effectiveness of physics-aware optimization in achieving the desired state for the kirigami, resulting in a significantly smaller deviation compared to the real deployed design without physics-aware optimization. This further underscores the importance of considering physics in the design process.

Reviewer #2 Comment #6: Is it necessary to manufacture such simple 2D kirigami structures by using advanced 3D printing techniques?

Response: We acknowledge the reviewer's observation that the geometry of kirigami itself can be accommodated by simpler manufacturing techniques. However, the primary challenge does not lie within the geometry but rather in achieving the precise magnetization orientations that are uniquely designed for all panels.

With the 3D printing in our study, magnetization programming can be realized in the printing process. With a permanent magnet attached around the dispensing nozzle, the polarities of magnetized particles in the ink are aligned to the printed direction during extrusion. The kirigami panel magnetization is thus programmed by controlling the filament printing direction. Through careful design of the printing path, we can ensure that the desired magnetization orientations of all panels are achieved upon completion of the printing process, without the need to do further processing.

In contrast, if we were to employ other methods to manufacture the magnetic kirigami geometry, precise programming of individual panel magnetization would be much more complicated. For example, one might consider manufacturing individual panels via molding and then magnetizing each separately. However, this approach would necessitate additional processing steps to connect the different panels together, which is not only complex but also prone to connection issues.

Therefore, based on these considerations, we believe 3D printing is a more appropriate choice to manufacture the design kirigami in this study, ensuring both manufacturing accuracy and efficiency. We have elaborated on the 3D printing process in greater detail in the Method section and Supplementary Information (SI).

Fig. S9. Direct ink writing printing process of magnetic kirigami. **a** Schematic direct ink writing printing of magnetic composite for a panel with programmed magnetization. **b** The optimized

kirigami design, the converted printing pattern, and the printed sample with the circular deployed state in Fig. 4.

Reviewer #2 Comment #7: I am wondering if the proposed design method can generate kirigami structures to undergo two successive but different deployed states under one applied magnetic field with different magnitudes?

Response: This is a very interesting question! Currently, the proposed method cannot design kirigami to precisely achieve two successive but different deployed shapes given the same direction but different magnitudes of external magnetic fields. The difficulty lies in addressing how to modify the optimization problem definition to simultaneously consider the physical and geometrical feasibility for both deployed states. We thought about three possible ways to realize this:

- 1) Define design variables only on one of the deployed states, use our energy model to find another deployed state under a different magnetic field, and impose geometrical constraints to ensure this deployed state meets the second target shapes.
- 2) Define design variables only on one of the deployed states, use our kinematic model to find the configurations of another deployed state, and aggregate energy gradients in both states into a single objective function to ensure physical feasibility.
- 3) Define two sets of design variables for the two respective deployed states and use the kinematic model to add extra geometrical constraints to ensure the two sets of variables correspond to the very same kirigami design.

Regarding the first option, it entails nested energy minimization within each optimization iteration, which compromises the differentiability of the proposed method and impedes the use of effective gradient-based methods. For the second option, using the kinematic model to determine the configuration in another state relies on knowledge of the deployed angle in that specific state, which is typically undetermined. Lastly, the third option substantially amplifies the dimensionality of the design space, increases the number of constraints, and introduces greater nonlinearity. This will make the optimization problem challenging to solve and highly sensitive to initial guesses. Consequently, our current method did not yield satisfactory designs for achieving this specific sequential shape-morphing behavior. However, we recognize it as a valuable avenue for future exploration, one that will demand more sophisticated modeling and optimization tools to effectively address.

Reviewer #2 Comment #8: There are many typo-errors in the manuscript; Figure 1 is easy to cause misunderstanding, as the shaded arrow looks like a design process with updated states; Figure 3 is hard to follow, as the readers may think the nodes are moved in different paths to evaluate the responses; the labels in subplots in figures are hard to follow when reading the main texts. The authors should revise the figure and the carefully examine the texts.

Response: We carefully proofread the manuscripts and corrected the typos.

Regarding Figure 1, we would like to clarify that all the components enclosed within the shaded arrow are indeed integral elements of the comprehensive workflow involved in the design process. To mitigate any potential misinterpretation, we have taken measures to enhance the clarity of

Figure 1. This includes a revision of the figure layout and the addition of explanatory text within the figure itself, which explicitly states that the mapping and optimization steps represent the two stages in our design process:

Fig. 1: Schematic diagram of the physics-aware differentiable design of kirigami.

a Compact state (left) of a regular quadrilateral kirigami and its deployed state (right) transformed from the regular compact kirigami following a geometrically feasible path. The repeated four-panel unit cell is shaded in blue, where the arrow shows the magnetization orientation of each panel. **b** Geometrical compatibility requirements for a four-panel cell in edges (equal-length pairs connected by red curved arrows) and angles around the center node (red sectors). **c** Torque induced by the applied magnetic field (blue curved arrow) and the mechanical forces/torques induced by deformation (red curved arrows) are required to be in equilibrium for a physically feasible deployed state. **d** The target deployed state (right) is conformally mapped from the deployed kirigami in Fig. 1a to achieve predetermined shapes (red dashed circular contour as an example) under the magnetic field \mathbf{B} . However, there is no compact state (left) that can follow a geometrically and physically feasible path to this mapped deployed state. **e** Compact state (left) and deployed state (right) of the kirigami optimized from the conformally mapped design in Fig. 1d, satisfying both physical equilibrium and geometrical compatibility. **f** This physics-aware optimization integrates two models, the differentiable kinematic model (top) and the differentiable energy model (bottom) under a fixed external magnetic field. The enlarged inset shows the equivalent nonlinear springs for kirigami hinges. The optimization process achieves physical equilibrium by ensuring minimal energy of the deployed kirigami indicated by zero gradients.

Regarding Figure 3, we acknowledge the concern that the abstract graph representations, particularly those arrows, could potentially lead to confusion. To address this issue, we have implemented several significant changes to Figure 3 in the revised manuscript:

- 1) We have streamlined the presentation by reducing the original full kirigami patterns to only their right halves.
- 2) In Figure 3a, we have removed the arrows and introduced a new subfigure (Figure 3b) to further illustrate the three basic geometry problems involved in the kinematic analysis of the two panels.
- 3) We have added shaded arrows to clarify the relationships among Figure 3a, Figure 3b, and Figure 3c. Essentially, we obtain the nodal locations in Figure 3a by solving the three basic geometry problems in Figure 3b. The analytical solution expressions derived in Figure 3b can be viewed as transformations from known nodes to nodes requiring calculation, forming the forward computational graph in Figure 3c.
- 4) The original Figure 3e has been divided into two subfigures: Figure 3e, which displays the kirigami configurations throughout the sequential solving process, and Figure 3f, positioned directly beneath it, which presents the corresponding computational graphs.
- 5) We also added explanatory text within the figure and modified the captions.

The modified figure and caption are given below:

Fig. 3: Differentiable kinematic model.

a Two exemplified panels in kinematic analysis corresponding to the red dashed box in the subfigure below. The grey dot is fixed and the locations of the remaining colored nodes are solved, which can be divided into three basic geometry problems indicated by shades in different colors. **b** Details of three basic geometry problems to be solved sequentially. In each problem, nodes in colors can be analytically solved based on the fixed nodes (grey) and already-solved nodes (white). The solution of each problem can be considered as a transformation from known nodes to nodes to be solved. **c** Extracted forward computational graph (left) from Fig. 3b by composing the transformation in sequence. Different types of transformations are marked by arrows in different colors, pointing from preceding nodes to the updated nodes. Shaded regions represent geometry problems, from which the analytical expression of the transformations f_i are obtained. A simplified graph is shown on the right. X_0 and θ are the original node locations and deployed angle, respectively. X_i is the set of updated nodes obtained from each transformation, corresponding to nodes of the same color in the original graph. **d** Backward computational graph to calculate the gradient of locations for the node of interest by composing the gradient of individual transformation ∂f_i via chain rules. **e** Sequential solving process to obtain all nodal locations. Only the right half is shown here as the left half is obtained by mirror symmetry on the center line. **f** Forward computational graph corresponding to sequential steps in Fig. 3e. The analytical transformations in each step are marked by arrows in different colors. Nodes that have been updated in previous steps are colored in white. **g** Backward computational graph (marked in red) to calculate the gradient of a given node.

For the description of Figure 3 in the main text, we adjusted the structures in Line 249 so that the writing flow matches with the label ordering in the figure:

“Given any change in the deployed angle, we solve for the corresponding kirigami configuration (kinematic analysis) sequentially as shown in Fig. 3. Specifically, calculating nodal locations of kirigami panels (Fig. 3a) involves a series of simple geometrical problems (marked by shaded regions in Fig. 3b), in which we solve for the unknown nodes (colored nodes) based on their constraints with the nodes obtained in preceding problems. Solutions for nodal locations in these basic problems can be analytically given, which can be viewed as a series of analytical transformations f_i (marked by colored arrows in Fig. 3c) from the preceding nodes to unsolved nodes (X_0 to X_3 and so on). By composing these transformations in sequence, they form a directed computational graph for forward analysis (Fig. 3c). Using chain rules, we can obtain the gradient values of node locations of any nodes by tracing the computational graph backward and multiplying the gradient of basic transformation ∂f_i in order (Fig. 3d). We apply this kinematic analysis to the whole kirigami, iterating nodes in every column of panels or voids (Fig. 3e) and sequentially composing the transformations together to get the analytical expression for the kirigami configuration. This analytical expression can be represented by a directed computational graph in Fig. 3f. Note that, due to the inherent regularity of cutting topology, i.e., the interconnections between panels, the solutions in each column only involve a limited set of transformation types (indicated by arrows in different colors in Fig. 3f). Using the chain rule again, we can readily calculate the analytical gradient of any nodal locations with respect to the deployed angle θ and initial node locations X_0 (Fig. 3g).”

References

R1 Reddy, J. N. *Theory and analysis of elastic plates and shells*. (CRC press, 2006).

- R2 Tao, J., Khosravi, H., Deshpande, V. & Li, S. Engineering by cuts: How kirigami principle enables unique mechanical properties and functionalities. *Advanced Science* **10**, 2204733 (2023).
- R3 Choi, G. P., Dudte, L. H. & Mahadevan, L. Programming shape using kirigami tessellations. *Nature materials* **18**, 999-1004 (2019).

REVIEWERS' COMMENTS

Reviewer #1 (Remarks to the Author):

The authors have addressed the reviewer comments very well. The paper is well-written and novel and it deserves to be published.

Reviewer #2 (Remarks to the Author):

I appreciate the authors' great efforts for replying my concerns. All my comments have been properly replied and the main texts have been improved. I believe the paper deserves positive acceptance in its current version in Nature Communications.

Response to the Reviewers' report for "Physics-aware differentiable design of magnetically actuated kirigami for shape morphing"

Reviewer #1 Comment #1: The authors have addressed the reviewer comments very well. The paper is well-written and novel and it deserves to be published.

Reviewer #2 Comment #1: I appreciate the authors' great efforts for replying my concerns. All my comments have been properly replied and the main texts have been improved. I believe the paper deserves positive acceptance in its current version in Nature Communications.

Response to Reviewer #1 & Reviewer #2: We would like to thank the reviewers for taking the time to thoroughly review our manuscript and provide constructive comments. The comments are insightful and very helpful for us to improve the paper's quality. We have made some small modifications to address editorial requests on formatting, including splitting the original Fig.6 into two smaller figures to meet the layout requirements. All the changes are highlighted in yellow in the updated manuscript.